# Time-integrated BMP signaling determines fate in a stem cell model for early human development

Seth Teague [1], Gillian Primavera[1], Bohan Chen [2], Zong-Yuan Liu[3], LiAng Yao[3], Emily Freeburne[3], Hina Khan[3], Kyoung Jo[3], Craig Johnson[3] & Idse Heemskerk [1,2,3,4,5] ✉

How paracrine signals are interpreted to yield multiple cell fate decisions in a dynamic context during human development in vivo and in vitro remains poorly understood. Here we report an automated tracking method to follow signaling histories linked to cell fate in large numbers of human pluripotent stem cells (hPSCs). Using an unbiased statistical approach, we discover that measured BMP signaling history correlates strongly with fate in individual cells. We find that BMP response in hPSCs varies more strongly in the duration of signaling than the level. However, both the level and duration of signaling activity control cell fate choices only by changing the time integral. Therefore, signaling duration and level are interchangeable in this context. In a stem cell model for patterning of the human embryo, we show that signaling histories predict the fate pattern and that the integral model correctly predicts changes in cell fate domains when signaling is perturbed. Our data suggest that mechanistically, BMP signaling is integrated by SOX2.

Secreted signaling molecules (morphogens) play key roles in cell fate decisions during embryonic development in vivo, as well as in stem cell models in vitro[1–5]. However, the relationship between morphogen signaling and cell fate patterning remains incompletely understood. It is generally accepted that the concentrations of signaling molecules determine gene expression and subsequent cell fate choices[6–9]. However, this model does not account for time: concentrations of signaling molecules and downstream signaling activity inevitably change as an embryo develops and cells are therefore unlikely to see a constant signaling environment for the duration of a particular cell fate decision (competence window). This problem is particularly acute in early mammalian development, where no maternal cues are present and signaling gradients are formed in feedback loops by the same cells that differentiate in response to them[4].

To understand how signaling controls cell fate one should therefore measure the full signaling history instead of focusing on a single point in time. This is technically challenging because differentiation takes place in a crowded, changing cellular environment and for mammalian cells can take multiple days[10–13]. In addition to the technical challenge, considering signaling histories raises a new conceptual problem. Rather than dealing with a static signaling level as the sole parameter, the signaling history in a cell has a formally infinite number of parameters, including rate of signal change, duration, and relative timing of different signals. Therefore, unbiased exploration by direct manipulation of these parameters is impractical. Instead, we pursue an indirect approach leveraging spontaneous heterogeneity in signaling activity.

BMP is a key morphogen with a conserved role in dorsoventral patterning across the Bilateria[5,14]. How BMP controls embryonic patterning has been extensively studied and yet remains controversial. For example, two recent studies in zebrafish came to different conclusions about whether expression domains of different BMP targets

[1]Department of Biomedical Engineering, University of Michigan, Ann Arbor, MI, USA. [2]Department of Computational Medicine and Bioinformatics, University of Michigan Medical School, Ann Arbor, MI, USA. [3]Department of Cell and Developmental Biology, University of Michigan Medical School, Ann Arbor, MI, USA. [4]Center for Cell Plasticity and Organ Design, University of Michigan Medical School, Ann Arbor, MI, USA. [5]Department of Physics, University of Michigan, Ann Arbor, MI, USA. ✉e-mail: iheemske@umich.edu

are consistent with a gradient-threshold model[15,16]. Current data fall short in at least two respects. First, previous studies do not account for signaling history. Although the BMP signaling gradient in zebrafish and other model systems is known to change over time[15,17], it remains unclear how this affects gene expression patterns. In other settings, dynamics of signaling gradients were essential in explaining final gene expression domains[18–20]. Second, studies to date have typically linked average signaling activity with average gene expression. Predicting the approximate fate boundaries along a single axis (such as the dorsal-ventral axis in zebrafish) provides only a few data points with uncertainty introduced by averaging over meaningful heterogeneity, such as patterns along the orthogonal axes or subpopulations of cells with qualitatively different signaling dynamics[21]. In contrast, relating signaling to fate in single cells provides thousands of data points in the same embryo, enabling more stringent tests of different models.

Upon BMP4 treatment, hPSCs with colony geometry that is precisely controlled using substrate micropatterning undergo self-organized spatial patterning into concentric rings of different fates that are specified during gastrulation in vivo. This makes micropatterned hPSCs a useful model for human gastrulation known as a 2D gastruloid[22]. Due to its reproducibility and high throughput, this system is ideal for quantitative studies of differentiation and has led to many insights into the mechanisms of mammalian gastrulation[23–25], some confirming previous findings in the mouse[26,27] and others later confirmed in the mouse[28,29] or exploring human-specific aspects of development[30].

Here we used micropatterned hPSCs as a model for early human development to test if and how fate choices in response to BMP might be quantitatively explained by signaling history. To this end we performed live-cell imaging of signaling activity followed by iterative immunofluorescence staining to relate signaling to fate in the same cells. We found that combined histories of BMP and Nodal signaling accurately predict cell fate patterns in micropatterned colonies. To limit the combinatorial effects of different pathways[27,30,31], we then created conditions to isolate the relationship between BMP signaling and fate. This simplified patterning to a binary decision between epiblast-like and amnion-like cell fates.

To test which features of BMP signaling histories most strongly correlate with fate and to establish causality, we complemented analysis of micropatterned hPSCs with experiments in standard culture conditions where increased heterogeneity can be leveraged to detect how signaling and cell fate are related. We performed automated tracking of signaling linked to cell fate in large numbers of individual hPSCs. Using an unbiased statistical approach, we showed that measured BMP signaling heterogeneity strongly correlates with cell fate heterogeneity at the single-cell level. We found that the initial and final levels of BMP signaling were relatively uniform across cell fates and conditions but that the duration of signaling varied strongly and correlated with cell fate heterogeneity. However, by direct manipulation of signaling level and duration we demonstrated that the level and duration cause differentiation only by changing the time integral of signaling. Thus, a lower level of signaling for a longer duration leads to similar differentiation as higher signaling for a shorter duration, and there is no absolute threshold in the duration or the level of signaling to achieve differentiation.

We then screened for genes that directly reflect the integral of signaling to determine the mechanism by which cells integrate signaling activity over time, which yielded SOX2 as a candidate among several other genes. We confirmed this using live imaging of endogenous SOX2 and constructed a simple mathematical model that accounts for all our data by assuming SOX2 represses differentiation genes and decreases in proportion to the time integral of BMP signaling. Finally, we confirmed a prediction of our model in which overexpression of SOX2 would reduce differentiation to amnion-like fate in response to BMP.

## Results

### Signaling dynamics in a stem cell model for human gastrulation predict fate pattern

BMP, Wnt, and Nodal function in a transcriptional hierarchy during self-organized pattern formation in micropatterned hPSCs[26,27] (Fig. 1A). Wnt and Nodal, as well as FGF signaling are required for primitive streak-like and primordial germ cell-like differentiation, whereas BMP alone is sufficient for amnion-like differentiation[22,26,27,30–32]. We previously measured the activity of the BMP and Nodal signaling pathways and found that SMAD4 signaling is dynamic, so static level thresholds cannot account for the final cell fate pattern[33]. Moreover, at the single-cell level, BMP signaling at the end of differentiation correlates poorly with cell fate, even under conditions where other signals are pharmacologically inhibited (Supplementary Fig. 1A, B). Here, we therefore asked if and how dynamic signaling could instead explain the cell fate pattern.

We began by live imaging hPSC colonies over 48 h of differentiation in cells expressing either GFP::SMAD4[34] or RFP::SMAD1[35] at the endogenous locus and quantified nuclear SMAD levels relative to cytoplasmic levels as a proxy for signaling activity (Fig. 1B–D). Although SMAD1 responds only to BMP, SMAD4 responds to both Nodal and BMP (Fig. 1B). To analyze spatiotemporal signaling patterns we exploited the approximate rotational symmetry of the system and averaged signaling over cells at the same distance from the colony edge (Fig. 1E, F). Consistent with previous work, GFP::SMAD4 signaling was initially uniform but became restricted to the edge around 12 hours with a wave of increased signaling starting around 24 hours[33]. RFP::SMAD1, which had not been measured during patterning before, matched GFP::SMAD4 except for the late signaling wave (Supplementary Fig. 1C, D), confirming our previous finding that this wave reflects Nodal activity, since RFP::SMAD1 does not respond to Nodal[2].

We then asked if unbiased data analysis could uncover structure in the radially averaged signaling histories. Principal component analysis on the SMAD4 signaling histories revealed a tripartite zigzag structure in the signaling histories (Fig. 1G) that we therefore computationally divided into three clusters (Fig. 1H, methods). Cluster means revealed that these represented cells in which signaling was always high (red), high then low (green), or high then low then high again (blue) (Supplementary Fig. 1E). The signaling clusters formed a spatially coherent pattern even though the clustering did not use spatial information (Fig. 1I). We then determined fate patterns in the same colonies by bleaching fluorescent proteins after live imaging and subsequently staining for fate markers in the same channels (Fig. 1J, Supplementary Fig. 1F). The resultant fate pattern closely resembled the pattern of signaling clusters (Supplementary Fig. 1G). We also computed the pattern for other numbers of signaling clusters but these did not divide the PCA plot into its obvious three parts and their biological relevance is unclear (Supplementary Fig. 1H).

We conclude that qualitatively different classes of signaling histories predict cell fate (in the statistical sense, which does not imply causality). Our computational approach thereby recovers previous qualitative observations that sustained BMP signaling leads to amnion-like differentiation while transient BMP followed by Nodal correlates with primitive streak-like differentiation and transient BMP without Nodal remains pluripotent[27,33,34]. Nevertheless, this result can be considered surprising for several reasons. First, enough information was provided by measurement of only one signaling protein that is part of two pathways (BMP, Nodal) out of at least four different pathways that are essential for pattern formation (also Wnt and FGF). Second, this result implies qualitative differences in signaling behavior between the fates: a smooth static signaling gradient would not allow the prediction of cell fate domains because it contains no information on where the downstream thresholds that determine fate boundaries are. Third, there

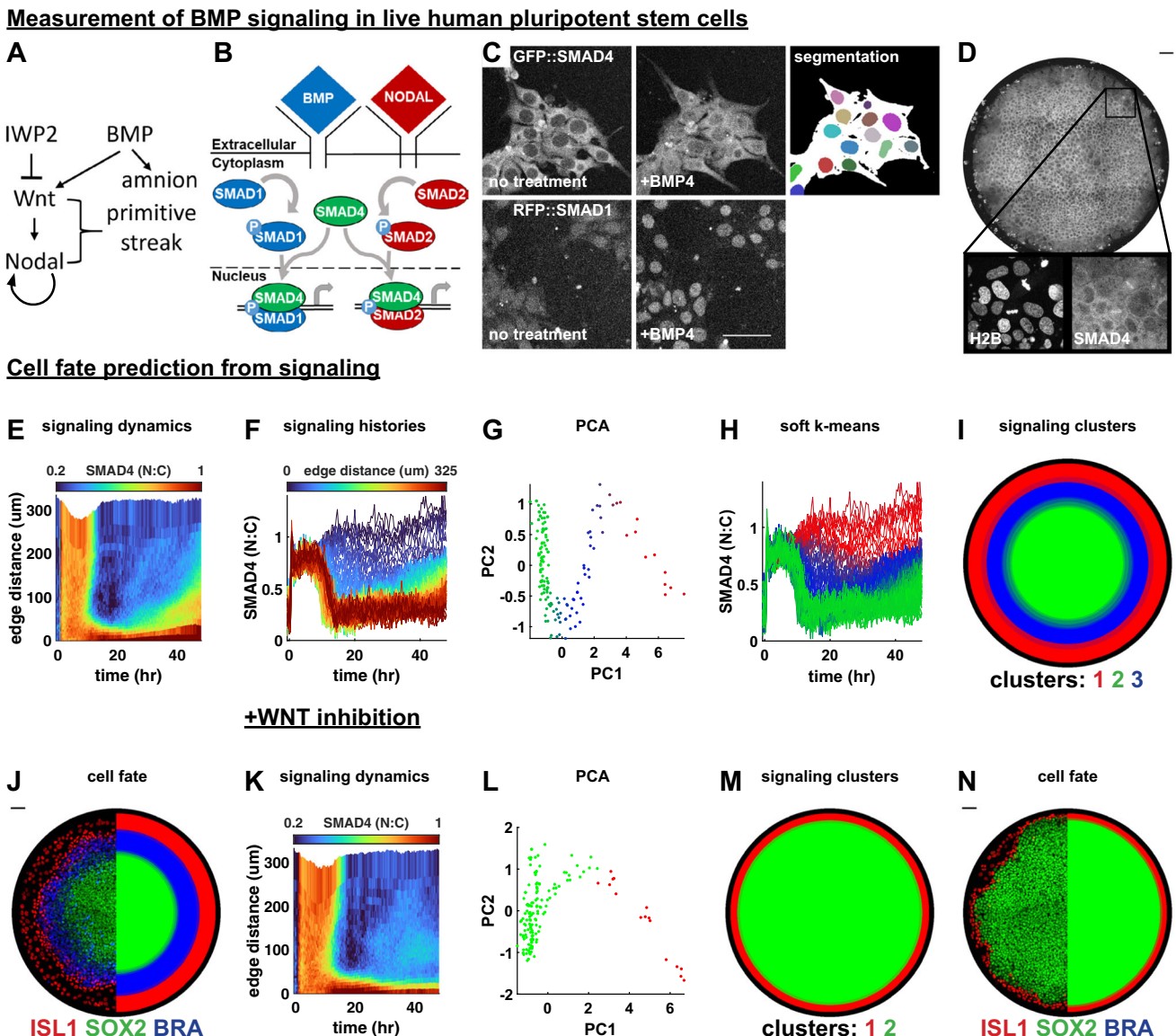

**Fig. 1 | Sgnaling dynamics in a stem cell model for human gastrulation predict fate pattern. A** Schematic of the BMP, Wnt, and Nodal signaling hierarchy and cell types induced by these signals. **B** SMAD1 translocates to the nucleus in response to BMP while SMAD4 translocates to the nucleus in response to both BMP and Nodal. **C** Nuclear translocation of fluorescently tagged SMAD4 (top) and SMAD1 (bottom) proteins in response to BMP4 treatment, and segmentation of nuclei (color) and cell bodies (white) in cells expressing GFP::SMAD4. **D** Micropatterned colony of RUES2 cells expressing GFP::SMAD4 at t = 30 hours after treatment with BMP4. **E** A heatmap of average spatiotemporal SMAD4 signaling dynamics (kymograph) in N = 5 micropatterned colonies treated with BMP4. **F** Plot of radially averaged signaling histories colored for distance from the colony edge. **G** Scatterplot of the first two principal components (PCs) of radially averaged signaling histories, colored for soft k means cluster assignment. **H** Plot of radially averaged signaling histories colored for cluster assignment. **I** Signaling clusters; each radial bin is assigned a color according to the dominant cluster of signaling histories within that bin, over N = 5 replicate colonies. **J** Immunofluorescence image of ISL1, SOX2, and BRA in a BMP4-treated colony (left) and the discretized fate map, averaged over replicate colonies (right). Each radial bin is colored for the dominant cell fate within that bin. **K** SMAD4 kymograph averaged over N = 5 replicate colonies treated with BMP4 and WNTi. **L** Scatterplot of the first two PCs of radially averaged signaling histories, colored for cluster assignment split 1 in Supplementary Fig. 1S (see methods). **M** Signaling clusters, created as in I. **N** IF image of a colony (left) and average fate map (right). Scale bars 50um. Source data are provided in a Source Data file.

have been claims that the initial state of a cell predicts its fate[36,37], which seems at odds with the signaling determining its fate unless the signaling response and initial state are correlated.

To further challenge our computational approach for predicting fate from signaling, we repeated the analysis after blocking Wnt secretion, which led to the absence of both endogenous Wnt and Nodal signaling due to the Wnt-Nodal hierarchy (Fig. 1A, Supplementary Fig. 1I). As expected, the late signaling wave in GFP::SMAD4 was eliminated (Fig. 1K), whereas RFP::SMAD1 was unaffected (Supplementary Fig. 1JK). SMAD2/3 staining showed a drop similar to SMAD4 (Supplementary Fig. 1L–N), consistent with the SMAD4 wave reflecting

Nodal signaling that is lost upon Wnt inhibition. PCA of SMAD4 signaling now yielded only two parts connected by an elbow (Fig. 1L), and clustering correctly predicted a binary fate pattern of amnion-like and pluripotent cells (Fig. 1MN, Supplementary Fig. 1O–S). The fact that the two signaling clusters are still connected after eliminating the middle cluster from Fig. 1G is explained by radial averaging. Eliminating primitive streak-like cells creates a boundary between amnion-like and pluripotent cells, where it leads to averaged signaling between these fates. This suggests that the signaling clusters would be better separated at the single-cell level. Overall, these results demonstrate that this approach correctly predicts cell fate patterns from

signaling and recapitulates known biology even after signaling disruption.

Although GFP::SMAD4 and RFP::SMAD1 looked qualitatively similar with Wnt inhibition, we noticed a difference in the time at which uniform BMP signaling is restricted to the edge (Fig. 1K, Supplementary Fig. 1J). However, RFP::SMAD1 and GFP::SMAD4 dynamics matched exactly (Supplementary Fig. 1T–W) when the two cell lines were mixed in the same colony. This suggests that the dynamics in (Fig. 1K, Supplementary Fig. 1J) reflect colony-level differences in excluding BMP from reaching the receptors[28], rather than distinct SMAD4 and SMAD1 dynamics downstream of BMP, so that both can be used interchangeably as readouts of BMP signaling when Wnt secretion is inhibited.

## A pipeline relating single-cell signaling history to fate

Having established that qualitatively distinct signaling histories match the cell fate pattern in micropatterned colonies, we asked if and how specific signaling features correlate with fate. To avoid the full complexity of dynamic combinatorial signaling, we focused on the decision between amnion-like and pluripotent cells controlled by BMP in the absence of WNT (and downstream Nodal). The analysis in Fig. 1 suggests that BMP response is high throughout differentiation in future amnion-like cells. However, it cannot be determined whether there is a minimum level or duration of response required for amnion-like differentiation since only a very small range of levels and durations are represented and each history is an average over a cell population. To address this problem, we therefore developed an experimental and computational pipeline to obtain cell signaling histories linked to fate in individual cells and applied this to a disorganized initial state (i.e., standard culture conditions) to leverage "spontaneous" heterogeneity and sample as broad a range of signaling responses as possible (Fig. 2, methods).

To obtain single-cell signaling histories we modified a broadly used automated tracking approach[38], in particular to better handle cell division (Fig. 2C, methods). Because tracking dense cells over multiple days is challenging, we labeled sparsely (10-20%) to obtain more reliable tracks. Manual verification yielded a linking accuracy of 98.8% between frames. To match signaling histories to fate we fixed cells after live imaging, immunofluorescence stained them, and reimaged the same positions. Images of nuclei from live and fixed data were then registered and cells were linked between images to obtain a data structure that contains gene expression data and signaling history for each cell, all in a fully automated manner (Fig. 2). Per experiment we could obtain several hundred to a thousand signaling histories linked to fate.

Because the cells express two different fluorescent proteins the number of cell fate markers that could be stained and imaged independently is reduced. As for the micropatterned colonies in Fig. 1, we therefore photobleached fluorescent proteins before staining[39], freeing up all channels. For some experiments we combined this with the 4i iterative immunofluorescence protocol to obtain multiple rounds of staining data[40]. Following previous work, we log-transformed and normalized expression data for cell fate markers to facilitate downstream analysis (see methods).

We optimized cell density and BMP dose for maximal cell fate heterogeneity (50% differentiation), which we expected to be most informative about the relationship between signaling and cell fate and obtained this at several densities with different doses of BMP4 (Supplementary Fig. 2A, B). We then evaluated SMAD4 signaling distribution over time at intermediate cell density (Supplementary Fig. 2C). Consistent with previous work, different concentrations of BMP yielded similar high initial signaling levels but differed in when signaling started decreasing[33,34]. BMP degradation provides an explanation for dose-dependent decrease[33] but not for saturation in duration of response and more rapid shutdown at high doses (Supplementary

Fig. 2C). However, we noticed that reduced BMP response at high doses coincided with colonies merging and cells reaching a high degree of confluence, in line with earlier work showing BMP response is restricted to colony borders[28] (Supplementary Fig. 2D). Consistently, media transferred from cells whose response to high BMP had dropped induced a strong response in untreated cells at lower density while adding fresh BMP to the original cells no longer elicited a response (Supplementary Fig. 2D, E). This suggests a combination of BMP degradation and receptor accessibility causes BMP signaling dynamics. It also explains why in micropatterned colonies signaling does not drop across all cells: micropatterning ensures that colony edges with high BMP response are maintained. However, we emphasize that this work focused on how signaling dynamics control fate, and to address this question the upstream regulation of signaling only matters insofar as it generates a wide enough range of dynamics to infer their relationship with fate. Consistent with signal interpretation being independent of upstream events, we observed throughout this work that similar BMP signaling under otherwise different conditions led to similar differentiation (e.g. shutdown from confluence versus small molecule inhibitors).

We repeated the experiments from Supplementary Fig. 2A–C with RFP::SMAD1 cells (Supplementary Fig. 2F–H), but differentiation was severely reduced, indicating that RFP::SMAD1 may be functionally compromised. However, consistent with Supplementary Fig. 1T–W, RFP::SMAD1 dynamics matched GFP::SMAD4, suggesting that SMAD1 localization is not affected and faithfully reports BMP response. Nevertheless, measurement noise in RFP::SMAD1 cells was greater than for GFP::SMAD4 (Supplementary Fig. 2I–K) and increasing light exposure to improve signal-to-noise led to phototoxicity. Therefore, we decided to focus on GFP::SMAD4 cells for relating signaling to differentiation and used RFP::SMAD1 as a control for average BMP signaling dynamics. To confirm that GFP::SMAD4 dynamics reflect only BMP signaling in the presence of Wnt inhibitor, we treated with the TGF-β receptor inhibitor SB431542 (TGFβRi), which reduced nuclear SMAD2/3 but had no effect on baseline GFP::SMAD4, suggesting GFP::SMAD4 does not respond to low constant levels of TGF-β signaling in base medium (Supplementary Fig. 2L, M). In contrast, Activin treatment induced a strong response in both SMAD4 and SMAD2/3. Staining for phospho-SMAD1/SMAD5/SMAD9 (pSMAD1/5/9) after BMP treatment strongly correlated with nuclear GFP::SMAD4 and correspondingly showed response to different doses of BMP that was initially similar but diverged later (Supplementary Fig. 2N, O). In summary, we established an automated pipeline to obtain signaling histories corresponding to high-dimensional cell fate data in large numbers of individual hPSCs and determined the optimal conditions to apply this pipeline to BMP signaling.

## The time-integral and duration of BMP signaling correlate with cell fate at the single-cell level

We tracked GFP::SMAD4 signaling histories for 42 h in the medium density condition with maximal heterogeneity and stained for seven transcription factors after live imaging: ISL1, GATA3, TFAP2C, and HAND1, which mark amnion-like cells and SOX2, NANOG, and OCT4 marking pluripotent cells. We first analyzed the structure of the signaling histories by themselves. Individual signaling histories varied around a sigmoidal mean (Fig. 3A, Supplementary Fig. 3A). To discern the dominant modes of variation between histories we again used principal component analysis. Visual inspection suggested an interpretation for the first three principal components (PCs) corresponding to duration, level of initial response and level of final response (Fig. 3B). To support this interpretation, we directly fitted a sigmoid curve to each signaling history to obtain these features (Fig. 3C) and correlated them with the PCs (Supplementary Fig. 3B). This confirmed PC2,3 respectively showed strongest correlation with high level and low

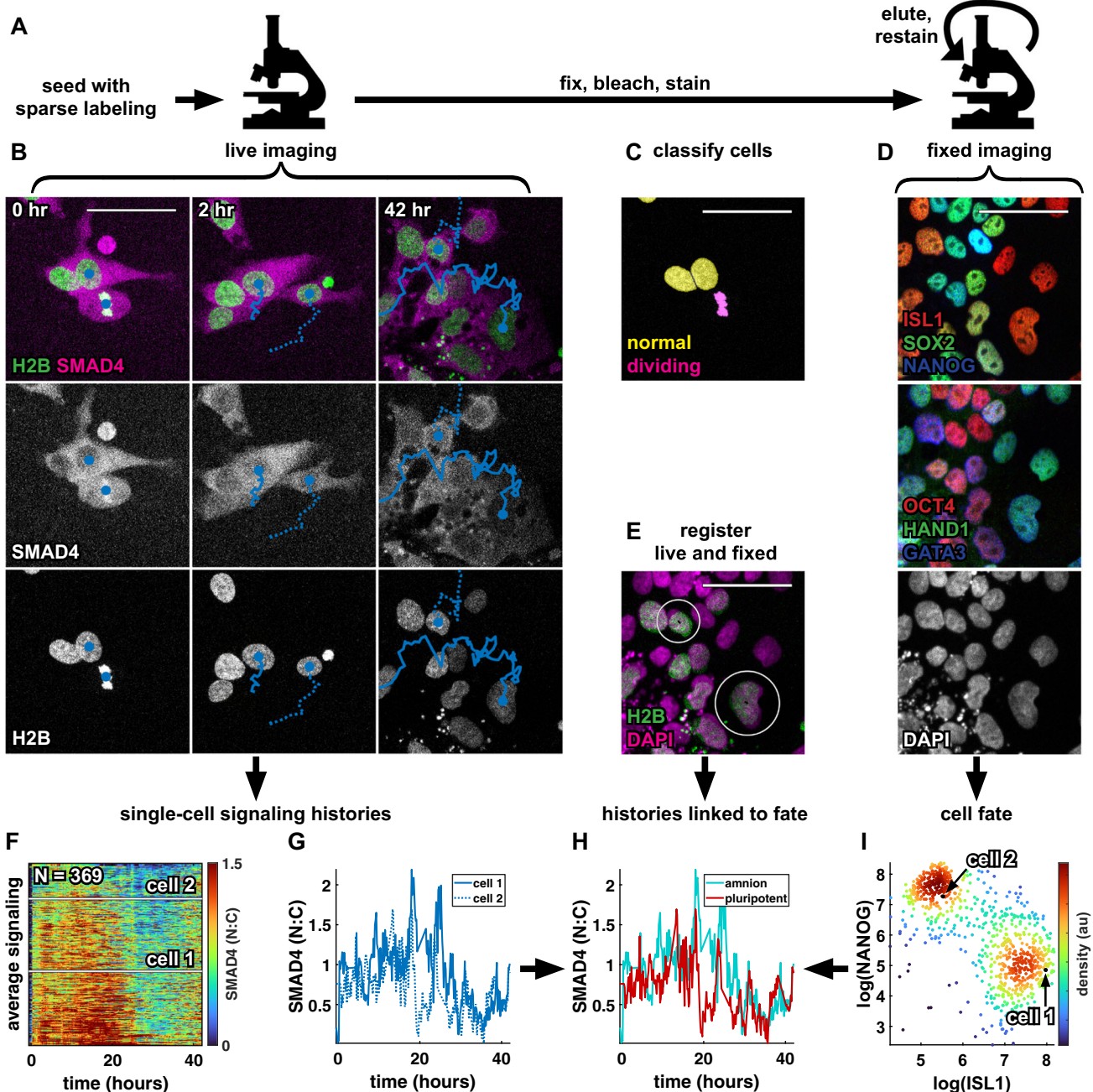

**Fig. 2 | A pipeline relating single-cell signaling history to fate. A** Schematic of the experimental procedure to collect data on signaling histories and cell fate. **B** Example images showing two tracked cells 0, 2, and 42 hours after treatment with BMP4 + WNTi in the SMAD4 and H2B channels. Nuclear centroids are marked with a solid blue circle, tracks are indicated by a solid and dashed line. **C** Nuclei with overlaid classification of cells as non-dividing (yellow) or dividing (magenta). **D** Iterative immunofluorescence (IF) data showing an initial stain for ISL1, SOX2, and NANOG (top), followed by a restain for OCT4, HAND1, and GATA3 (middle), in the same field of view as the live data in **B**. Bottom panel shows the DAPI image from the first round of fixed imaging. **E** Two-color overlay of the live (sparsely labeled) H2B image at 42 hours in **B** (green) and the DAPI image in **D** (magenta). The two cells for which tracks are shown in **B** are circled in white. **F** Heatmap of all single-cell signaling histories collected in a single experiment (*N* = 369 histories), sorted by mean signaling level. Signaling histories of the two cells tracked in **B** are marked by white lines. **G** Signaling histories of the two cells tracked in **B**. **H** Signaling histories of the two cells tracked in **B**, colored for fate after matching nuclei from the live imaging time lapse to nuclei in the fixed IF data. **I** Scatterplot of log-transformed ISL1 and NANOG intensity in single cells, colored for local density, showing a bimodal distribution of ISL1+ amnion-like cells and NANOG+ pluripotent cells. Scale bar 50um. Source data are provided in a Source Data file.

level. However, PC1 correlated strongly with all three signaling features.

The scatterplots of fit parameters versus PC1 contained outliers corresponding to histories with poor fitting of the sigmoid due to noise, to which the duration was most sensitive (Fig. 3D, Supplementary Fig. 3C,D). Signaling histories share with single-cell RNA-sequencing data that they are noisy high-dimensional single-cell

measurements. We therefore tested if we could effectively reduce noise in these signaling histories by data diffusion using MAGIC, an algorithm developed for single-cell RNA-sequencing data[41]. Weighted averaging of signaling between histories that are most similar using data diffusion visibly reduced noise in signaling histories (Fig. 3E), after which fits improved (Supplementary Fig. 3C, D) and outliers disappeared (Fig. 3D), making the relationships between fitted signaling

## BMP signaling histories vary around a sigmoidal mean

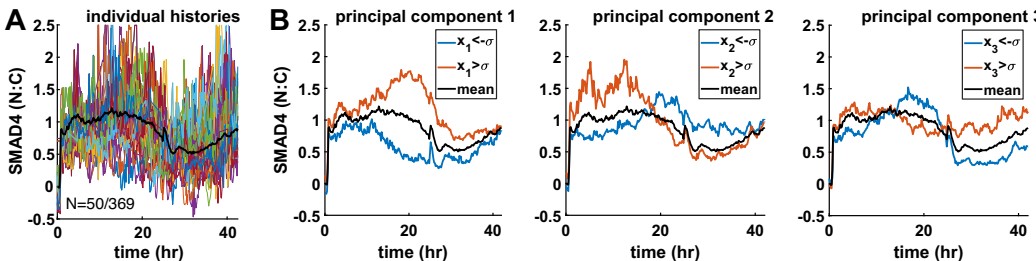

## The first principal component correlates strongly with the time integral of signaling

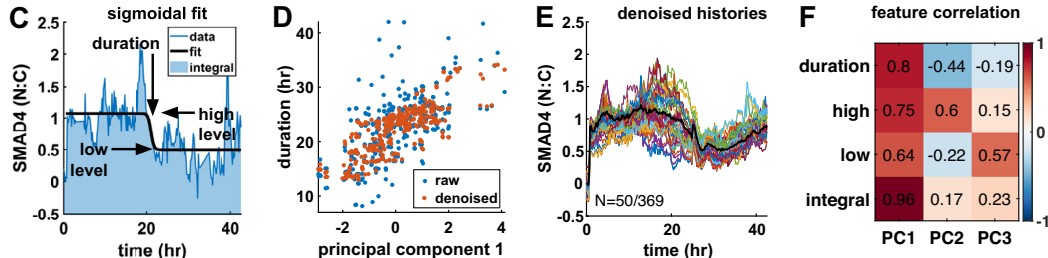

## Cell fate is best separated by ISL1/NANOG and correlates with signaling integral and duration

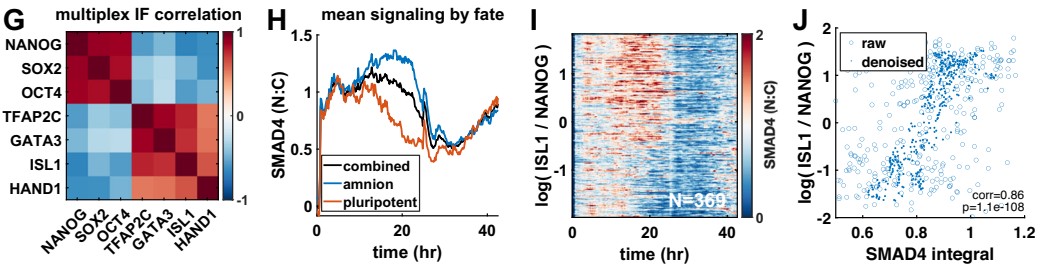

## The relation between signaling and fate is consistent across BMP doses and cell densities

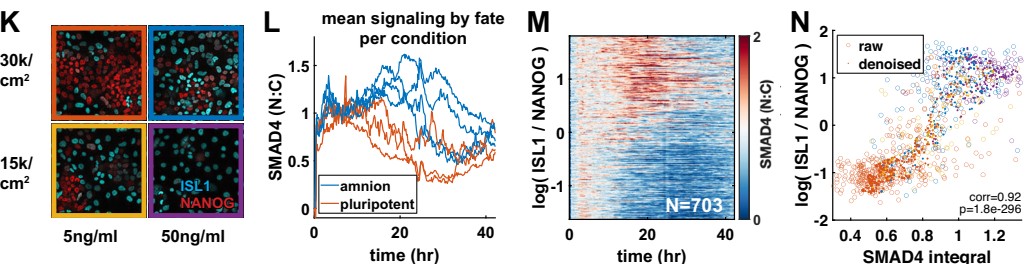

## The complete signaling history contains no more information about fate than the integral

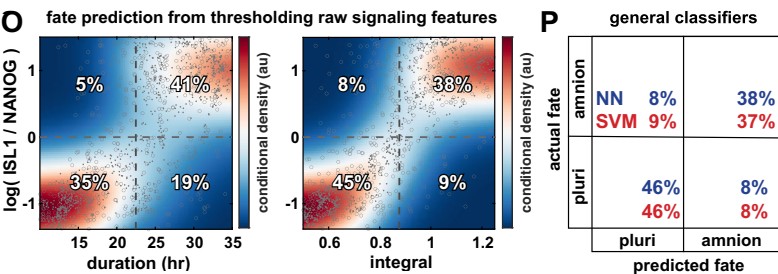

features and principal components more apparent. After denoising the total variance explained by the leading three principal components went from 48% to 83% with sub-leading components making very small contributions, suggesting that these sub-leading components mostly capture noise, consistent with the fact that they had no obvious

interpretation (Supplementary Fig. 3E). Although denoising increased the correlation between duration and PC1 relative to other features (Fig. 3F), the correlation was still high between all parameters and PC1. This suggested that PC1 may represent the time integral (i.e., total amount) of signaling, which increases with any one of the parameters.

**Fig. 3 | The time-integral and duration of BMP signaling correlate with cell fate at the single-cell level. A** Plot of 50 out of 369 signaling histories, population mean overlaid as a bold black line. **B** Separation of signaling histories along the first three principal components (PCs). Average of histories one standard deviation above or below the mean along each component are shown in red and blue, respectively. **C** Example signaling history with a sigmoidal fit to determine signaling features. **D** Scatterplot of duration vs. PC1 with and without denoising using MAGIC. **E** Signaling histories from **A** after denoising. **F** Correlation between signaling features and PCs across cells. **G** Correlation of fate marker expression across cells. **H** Mean signaling histories for amnion-like and pluripotent cells. **I** Heatmap of signaling histories sorted by log(ISL1 / NANOG). **J** Scatterplot of log(ISL1 / NANOG) against signaling integral for single cells, with and without denoising. *P* value calculated with the t-statistic in the MATLAB function corrcoef. **K** antibody stains for ISL1 and NANOG at different cell densities and BMP4 concentrations. **L** Mean

signaling for amnion and pluripotent fate for each condition in **K**. **M** Heatmap of signaling histories sorted by log(ISL1 / NANOG). **N** Scatterplot of log(ISL1 / NANOG) against signaling integral for single cells, with and without denoising. Color is by condition, indicated with color borders around images in **K**. *P* value is as in **J**. **O** Heatmap of kernel density estimate after denoising of conditional distributions of log(ISL1 / NANOG) with respect to duration and signaling integral overlaid with a scatterplots of data points before (circles) and after denoising (dots). Dashed lines show separation of cells into amnion-like and pluripotent based on log(ISL1 / NANOG) or on signaling features. The percentage of cells in each quadrant is indicated, with correct assignments in the top right and bottom left quadrant of each heatmap. **P** Confusion matrix showing the performance of a neural network (NN) and a support vector machine (SVM) in classifying cells as amnion-like or pluripotent using the full signaling history. Source data are provided in a Source Data file.

The resemblance of PC1 to duration in Fig. 3B is explained by differences in duration dominating variation in the integral. Indeed, we found that the integral of signaling correlates much more strongly with PC1 than other features (Fig. 3F, Supplementary Fig. 3F).

Having discovered duration, high level, low level, and integral as the key features of a signaling history, with integral as the dominant mode of variation, we asked whether these signaling features correlate with cell fate. We first looked at the fate data alone. As expected, the markers separated into two groups representing amnion-like and pluripotent cells (Fig. 3G, Supplementary Fig. 3G, H). Based on the seven-dimensional cell fate data we then clustered the cells into two discrete fates: amnion-like or pluripotent (differentiated or undifferentiated) (Supplementary Fig. 3H–K, methods), and calculated the average signaling history in each cluster. Strikingly, we found that on average the two fates have nearly identical initial and final (high and low) signaling levels but they differ significantly in their mean duration and integral (Fig. 3H). We confirmed this result with RFP::SMAD1 (Supplementary Fig. 3L), although SMAD1 data was much noisier. These key findings show that the dominant mode of variation in BMP signaling, the integral, correlates with the fate.

We then asked if signaling history is not only different between discrete cell fates but whether it correlates with expression levels of fate markers, that can be interpreted as coordinates along a differentiation trajectory[42]. We found that log expression ratio of ISL1 and NANOG, log(ISL1/NANOG), best separates the fates (Supplementary Fig. 3H–K, methods). We therefore focused on this single continuous variable along the fate trajectory for further analysis. A heatmap of signaling histories versus this continuous fate variable showed a clear pattern of higher and longer signaling for more differentiated cells (Fig. 3I). To reveal the relationship between specific signaling features and the degree of differentiation more clearly, we again denoised, with controls for artifacts (see methods). However, regardless of denoising, we found stronger correlation between fate and signaling duration or integral than between fate and high or low level (Fig. 3J, Supplementary Fig. 3M), consistent with the means in Fig. 3H.

We next asked if the relationship between signaling and fate remains fixed when their distributions are changed by varying cell density and BMP dose. This would be expected in simple models relating signaling and fate, although a more complex context-dependent relationship is possible. We indeed found a consistent relationship between signaling and fate across two different densities and two different BMP doses (Fig. 3K). Mean signaling for amnion-like cells in any condition was separated from mean signaling for pluripotent cells in any condition (Fig. 3L, Supplementary Fig. 3N), and the heatmap of combined histories versus fate showed a consistent trend that was much more pronounced than for a single condition (Fig. 3M). Finally, different conditions combined to form a clear threshold-like relationship between signaling features and fate with plateaus for undifferentiated and fully differentiated cells (Fig. 3N, Supplementary Fig. 3O). Importantly, consistent with (Supplementary Fig. 2C), varying

BMP concentration and cell density changed the duration of the response but showed minimal variation in the initial and final signaling levels (Fig. 3L).

Although the integral showed the highest correlation with fate, all features correlated strongly with fate and with each other. We therefore asked if all information about fate is contained in the signaling integral, or whether different features contain independent information. To test this, we first calculated how well an optimal threshold for each feature separates the fates. The percentages of cells in the quadrants formed by the cell fate and signaling feature thresholds in Fig. 3O then provide the confusion matrix of the Bayesian classifier (see methods). The upper right and left quadrants respectively correspond to correctly predicted amnion and amnion misclassified as pluripotent based on the feature threshold, while the lower left and right correspond to correctly predicted pluripotency and pluripotent cells misclassified as amnion. We found the integral-based prediction to be most accurate at 83% (Fig. 3O, Supplementary Fig. 3P). To estimate the total information contained in signaling we trained a variety of general classifiers, specifically artificial neural networks and support vector machines, and found their accuracies in predicting fate from the complete signaling history to be extremely similar, also around 83% (Fig. 3P). This suggests that the complete signaling history contains no more information about fate than the integral. Although we conservatively performed this analysis with raw data to exclude the possibility of denoising artifacts, we found that after denoising, integral-based prediction becomes nearly perfect (97% accuracy), further supporting the conclusion that all information about fate is contained in the integral.

In summary, we found that the signaling response to BMP is characterized by a high initial plateau going down to a lower final plateau. Although it is primarily the duration that varies between cell fates and experimental conditions, fate is best explained by the time integral of signaling, which in turn depends on the duration. This raises the question of whether mechanistically either the time-integral or the signaling duration controls fate.

## The time-integral of BMP signaling controls differentiation

Having found that both the duration and time-integral of signaling strongly correlate with fate, we asked how integral-dependent differentiation can be distinguished from a level threshold combined with a duration threshold, which is the simplest dynamic extension of the classic morphogen model. We reasoned that if the integral controls cell fate, lower levels of signaling can be compensated by a longer duration, which is inconsistent with an absolute level threshold (Fig. 4A–C).

To independently control signaling level and duration and determine how their combination impacts fate, we could not use treatment with different concentrations of BMP4, since these do not yield different steady levels of response[31,33] (Supplementary Fig. 2C). Therefore, we first combined low cell density with high BMP

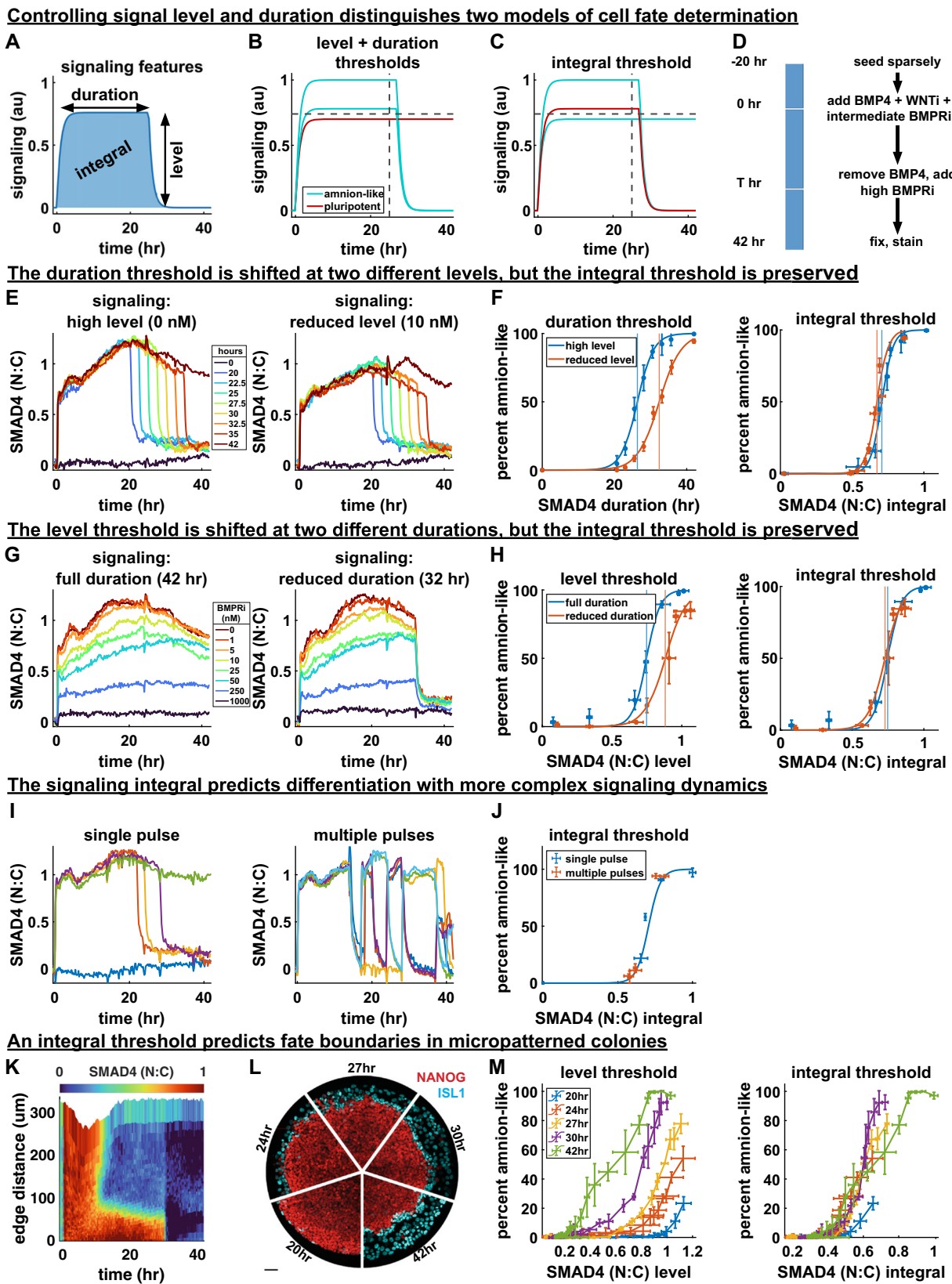

## Controlling signal level and duration distinguishes two models of cell fate determination

## The duration threshold is shifted at two different levels, but the integral threshold is preserved

## The level threshold is shifted at two different durations, but the integral threshold is preserved

## The signaling integral predicts differentiation with more complex signaling dynamics

## An integral threshold predicts fate boundaries in micropatterned colonies

concentration to obtain a high BMP response in all cells throughout differentiation. We then controlled the SMAD4 signaling level by treatment with different doses of the BMP receptor inhibitor LDN193189 (BMPRi) and the duration by BMP removal combined with a high dose of BMPRi to abruptly and completely shut down signaling (Fig. 4D, Supplementary Fig. 4A). As before, pSMAD1 and RFP::SMAD1

response matched GFP::SMAD4 (Supplementary Fig. 4B, C). Therefore, the signaling level was experimentally defined as the mean SMAD4 signaling before signaling shutdown and the duration as the time of the shutdown.

We first varied duration while holding level fixed (Fig. 4E). Consistent with our findings in Fig. 3 we found a sharp duration threshold

**Fig. 4 | The time-integral of BMP signaling controls differentiation. A** Diagram of relevant signaling history features. **B–C** Hypothetical set of signaling histories for which a combined level and duration threshold model makes a different prediction of cell fate than an integral threshold model. **D** Schematic of the experimental procedure used to control the level and duration of signaling. **E** (left) Mean signaling for different durations without initial BMPRi. (right) Mean signaling for different duration with an initial BMPRi treatment of 10 nM. **F** Duration and SMAD4 signaling integral thresholds based on logistic sigmoid fit for two signaling levels. Data in **F, H, J** are presented as mean +/− standard deviation (SD) over $N = 4$, 3, and 3 images, respectively. Thresholds in any signaling feature are defined by 50% differentiation. **G** Mean signaling for 8 doses of BMP inhibition with LDN193189 for durations of 42 hr (left) or 32 hr (right). **H** SMAD4 signaling level and integral thresholds for both signaling durations. **I** Mean signaling for single pulse controls (left) compared to mean signaling for multiple pulses of BMP signaling (right). **J** Differentiation vs. integrated signaling for one vs. multiple pulses. **K** Kymograph of average BMP signaling for $N = 3$ colonies treated with 200 ng/mL BMP4 in the presence of WNT inhibitor, treated with BMP inhibitor at 30 hours. **L** IF data showing amnion differentiation for each signaling duration (scale bar 50um). **M** Percent differentiation against mean signaling level before shutdown for each duration (left) and against signaling integral (right). Each point represents a radial bin (a fixed distance from the colony edge). Error bars are standard deviation over $N = 3$ colonies per condition. Source data are provided in a Source Data file.

around 26 hours, above which cells predominantly differentiated to amnion-like fate (Fig. 4F, Supplementary Fig. 4D). We then repeated this while lowering the level with a small dose of BMPRi throughout the experiment and found that the duration threshold went up to around 32 hours (Fig. 4E, F and Supplementary Fig. 4D), consistent with our hypothesis that there are no fixed level and duration thresholds. Moreover, when we plotted the duration-differentiation curves for two different levels against the time-integral of signaling, these collapsed on top of each other with an identical integral threshold (Fig. 4F). Of note, the duration threshold in Fig. 4F is slightly later than that found in Fig. 3, which is also consistent with integral control of fate because the final signaling level after BMPRi treatment is lower than in cells where the BMP response goes down spontaneously. Overall, the fact that the duration required for differentiation changes at different signaling levels but the integrated signaling remains the same provides strong quantitative support for the integral model.

To further support our integral hypothesis, we then explored a wide range of signaling levels while holding the duration fixed (Fig. 4G). We found a sharp threshold in level above which pluripotency was lost that was shifted upward when the duration of signaling was decreased (Fig. 4H). These level-differentiation curves again collapsed on a common integral threshold (Fig. 4H). We confirmed the relationship between integrated signaling and differentiation for pSMAD1 by fixing and staining for pSMAD1 at regular intervals (Supplementary Fig. 4E, F). We then tested whether the relationship between integrated BMP signaling and amnion-like differentiation depends on TGF-β/Nodal signaling by repeating the experiment in the presence of TGFβRi (Supplementary Fig. 4G, H). With TGFβRi, SMAD4 response to BMP4 with different doses of BMPRi was similar, although slightly stronger, consistent with literature describing an inhibitory effect of pSMAD3 on pSMAD1/5/9-SMAD4 complexes[43]. As expected, TGF-β inhibition caused downregulation of NANOG but left SOX2 unaffected[44] (Supplementary Fig. 4I). We therefore used log(ISL1/SOX2) to define fate and again found that differentiation to ISL1+ amnion-like cells was predicted by the integral of SMAD4/BMP signaling (Supplementary Fig. 4J). We also found that integrated signaling correctly predicted differentiation for multiple pulses of BMP signaling (Fig. 4IJ).

Finally, we asked if the integral model can account for differentiation in the micropatterned model for embryonic patterning, where a BMP signaling gradient forms spontaneously from the edge inward due to receptor accessibility and secretion of inhibitors[28,30] (Supplementary Fig. 4K). If BMP signaling is shut down across the colony, cells at different distances from the edge will have been at different signaling levels for the same duration and therefore have experienced different amounts of integrated signaling (Supplementary Fig. 4L). By shutting down BMP signaling in micropatterned colonies at different times and measuring differentiation as a function of distance from the colony edge we are then simultaneously testing the effect of level and duration. We therefore live-imaged micropatterned colonies of GFP::SMAD4 hPSCs with BMP4+WNTi, treated them with a high dose of BMPRi at different times and then fixed them after 42 h to evaluate the percentage of amnion-like cells at different

distances from the edge (Fig. 4K, L, Supplementary Fig. 4M-T). Consistent with the experiments in sparse culture, we found that the signaling level required for 50% differentiation was strongly dependent on the duration of signaling, but that level-differentiation curves approximately collapsed on a common integral threshold (Fig. 4M).

It has been claimed that GATA3 acts as an irreversible switch driving commitment to differentiation after as little as one hour of BMP signaling[45], which seems inconsistent with our findings. For direct comparison we therefore measured GATA3 in the same experiment (Supplementary Fig. 4V). We found that, similar to ISL1, it reflects the integrated signaling, but surprisingly does not show a switch-like threshold, appearing graded instead (Supplementary Fig. 4W, X). We therefore found no evidence of a BMP signaling duration threshold for GATA3 activation, early or late. Altogether, our data provide strong evidence that amnion-like differentiation is controlled by the time-integral of BMP signaling in both standard culture and micropatterned colonies and that there are no absolute thresholds in level of duration or level of signaling.

## BMP signaling may be integrated by SOX2
We asked by what mechanism cells integrate BMP signaling and reasoned that the simplest mechanism would be a protein that increases or decreases at a rate that is proportional to the level of BMP signaling on a timescale comparable to that of differentiation. A threshold response of differentiation markers like ISL1 or HAND1 downstream of such an integrator gene would then explain the integral threshold observed in Fig. 4. In the simplest case, the integrator would be a transcription factor directly regulating the downstream genes. We therefore looked for transcription factors showing the gradual increase or decrease on the timescale of differentiation with immediate response at a rate roughly proportional to the level of BMP signaling.

First, we measured protein-level dynamics of amnion and pluripotency genes for different levels of BMP signaling using immunofluorescence staining on a time series of fixed samples. We observed three classes of dynamics (Fig. 5A, D, Supplementary Fig. 5A): gradual increase with a dose-dependent slope (GATA3, TFAP2C), gradual decrease with a dose-dependent slope (SOX2, NANOG), and delayed increase (ISL1, HAND1). For genes showing immediate response we related their level to the integral of SMAD4 signaling for the same dose of BMPRi (Fig. 5B, Supplementary Fig. 5B) in the first 24 h and found an approximately linear relationship for each (Fig. 5C, Supplementary Fig. 5C). In contrast, for ISL1, HAND1, and OCT4 we confirmed the threshold dependence on SMAD4 of Fig. 4 (Supplementary Fig. 5D). Therefore, this analysis identified four of these seven genes as potential integrators (GATA3, TFAP2C, SOX2, NANOG).

We then performed similar analysis at the transcriptional level in a genome-wide, unbiased manner. We screened for integrator genes with bulk RNA-seq at different times after BMP treatment and at different levels of BMP signaling after 5 h. To focus on genes with large expression changes we restricted our analysis to genes with a cumulative fold change over time of at least one standard deviation above the mean (Supplementary Fig. 5E). Consistent with the

## SOX2 is part of a class of potential integrator genes

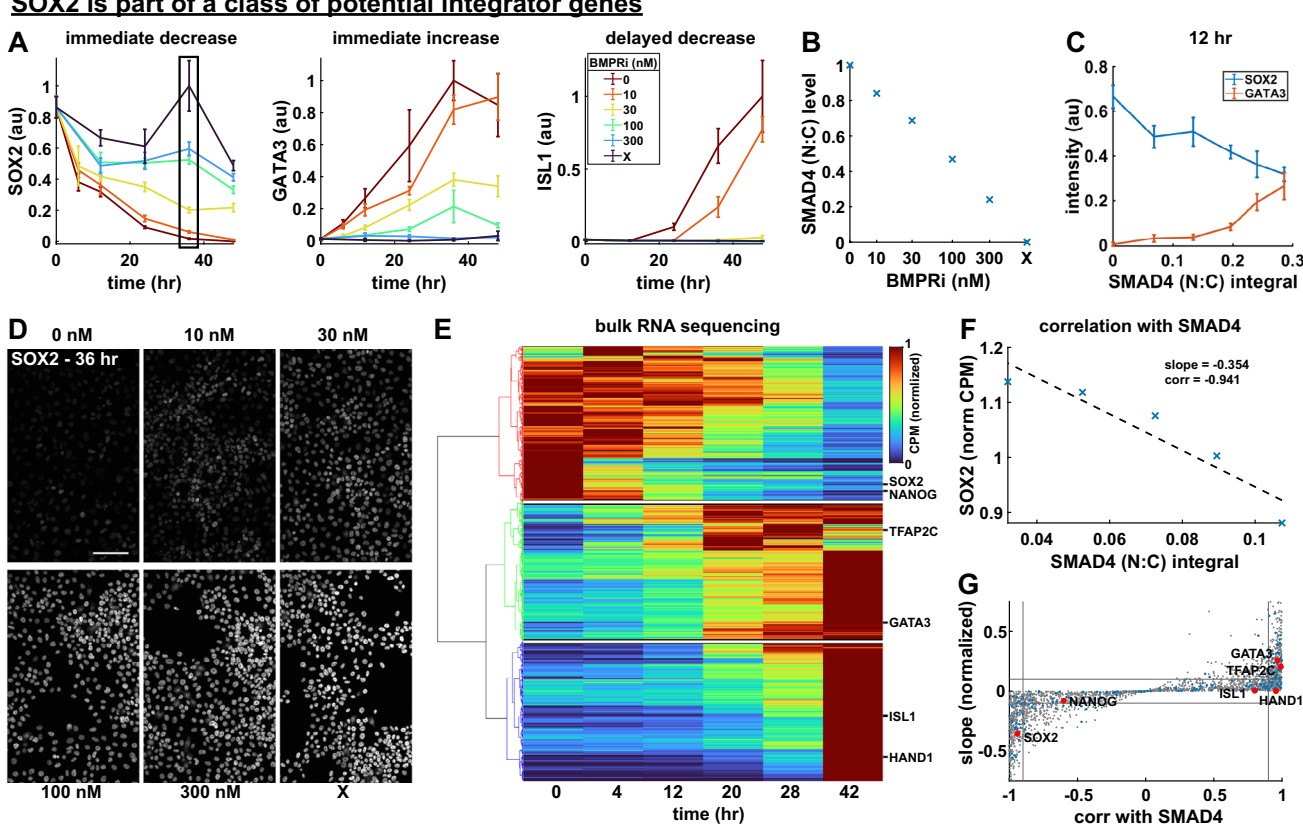

**Fig. 5 | BMP signaling may be integrated by SOX2. A** Normalized expression of SOX2, GATA3, and ISL1 over time for different signaling levels, measured with time-series IF. Data presented as mean +/− standard deviation (SD) across N = 6 images. A box outlined in black shows the data points corresponding to image data in **D**. **B** Average nuclear SMAD4 signaling level for each treatment condition in **A**, determined using SMAD4 dynamics measured in the same conditions and shown in Supplementary Fig. 5B. **C** Cross section of **A** showing SOX2 and GATA3 expression at 12 hours, plotted against SMAD4 signaling integral. **D** Example IF image data for SOX2 in each treatment condition at 36 hours, corresponding to the points boxed in **A**. Scale bar 50um. **E** Heatmap of time-series bulk RNA seq data (normalized counts per million) with genes on the y-axis ordered by hierarchical

clustering. The cluster dendrogram is shown to the left, with lines colored for discrete cluster assignment, and white lines are drawn on the heatmap to separate clusters. The location in the heatmap of genes also measured with IF is indicated to the right. **F** Example bulk RNA seq dose-response data is shown for SOX2 with a linear least squares fit of SOX2 with respect to SMAD4 signaling level. The slope of the least squares fit and the correlation coefficient between SOX2 and SMAD4 signaling level are indicated. **G** Scatterplot of the slope of each gene with respect to SMAD4 signaling and that gene's correlation with SMAD4 signaling in the dose-response bulk RNA seq data, as determined in **F**. Locations in the scatterplot of genes for which we also have IF data are indicated. Transcription factors are marked in blue, other transcripts in gray. Source data are provided in a Source Data file.

immunofluorescence data, hierarchical clustering of the time series then revealed three clusters of genes undergoing large changes in expression during differentiation: 816 continuously decreasing genes, 729 immediately increasing genes, and 696 delayed increasing genes, of which 79, 75, and 59 were transcription factors, respectively[46] (Fig. 5E, Supplementary Fig. 5F, Supplementary Data 1). We then identified genes with an immediate response to BMP that was both proportional and strong, by respectively filtering based on the correlation with SMAD4 signaling level (above 0.9) and slope (above 0.1 normalized to the time-series maximum) in the dose-response data after 5 h (Fig. 5F, G). Intersecting these with the set of genes from the time series analysis left 28 immediately decreasing and 38 immediately increasing transcription factors as candidate integrators. Reassuringly, these respectively contained SOX2 and GATA3, TFAP2C. However, NANOG was excluded due to low correlation with SMAD4 signaling at 5 h. This suggests NANOG is not a direct transcriptional target of BMP signaling and its response on longer timescales measured with IF is either indirect or post-transcriptional. Overall, the bulk RNA-seq data identified potential integrator genes and provides a deep characterization of the transcriptional response to BMP4 in hPSCs.

Decreasing genes are associated with the pluripotent state and increasing genes with amnion-like fate, raising the question of whether

the integral threshold represents a loss of pluripotency or a commitment to amnion fate. These are indistinguishable in our experiments because we excluded other fates with Wnt inhibition. However, we found that ISL1 and HAND1 anticorrelate more strongly with SOX2 and NANOG than they correlate with GATA3 and TFAP2C (Fig. 3G), suggesting that loss of pluripotency genes may be more important than gain of expression of early amnion genes to drive expression of late amnion genes. Furthermore, it was recently proposed that there is a time window in which cells expressing amnion markers can still acquire primitive streak-like fate by exposure to Wnt[31]. This also suggests that our integral threshold represents a commitment to differentiate, i.e. loss of pluripotency, rather than commitment to amnion-like fate. To single out a specific candidate integrator gene we therefore decided to focus on pluripotency genes. We then identified SOX2 as the most likely candidate since it was the only of the so-called 'core pluripotency genes' that fit the criteria for an integrator in the bulk RNA-seq analysis.

To test how well SOX2 levels reflect integrated SMAD4 signaling, we again varied levels of BMP signaling for two different durations as in Fig. 4, but now measured SOX2 over time using a cell line expressing GFP::SOX2 in the endogenous locus (Supplementary Fig. 5G, H). With SMAD4 signaling inferred from GFP::SMAD4 under the same

conditions (Fig. 4, Supplementary Fig. 5G), GFP::SOX2 dynamics during the first 16 hours of differentiation confirmed the linear relationship between SMAD4 signaling integral and the rate of SOX2 decrease over a wide range of SMAD4 signaling levels, as expected if SOX2 integrates SMAD4 signaling (Fig. 6AB) and consistent with similar observations by Camacho-Aguilar et al. [31].

Our data also showed deviations from the linear relationship at later times, in particular recovery after shutdown of BMP signaling, likely due to protein turnover (Supplementary Fig. 5H). We asked whether a simple model with production and degradation of SOX2 leading to exponential time-dependence could explain these deviations and still be consistent with our measured integral threshold. To answer this question, we implemented this model mathematically. We modelled SOX2 production as decreasing linearly with SMAD4 and decay as constant (Supplementary Fig. 5H), which resulted in SOX2 levels reflecting a weighted integral of BMP signaling that approximates the true integral for slow enough turnover (Supplementary Note 1). The simple exponential model showed generally good agreement with the observed SOX2 dynamics (Supplementary Fig. 5H). However, for high levels of BMP signaling, SOX2 did not plateau as expected and average SOX2 recovery rates were lower than expected after BMP shutdown at 32 h. We reasoned that SOX2 becomes permanently repressed and therefore does not recover in differentiated cells, which constitute a larger fraction after exposure to higher levels of BMP signaling. We modeled this at the level of the population means by adding negative regulation of SOX2 by ISL1, which resolved the observed discrepancies (Supplementary Note 1).

Our fitted model predicted a SOX2 half-life around 18 h (Supplementary Note 1). To test this we measured GFP::SOX2 fluorescence recovery after photobleaching (FRAP) and obtained a half-life of 7 h (Supplementary Fig. 5I–K), which is inconsistent with simple production/degradation to explain the measured SOX2 dynamics. It is well known that SOX2 regulates its own expression[47] and that response can be slowed down by positive autoregulation[48]. After including positive SOX2 autoregulation in our model, we were able to fit the observed SOX2 dynamics using the measured SOX2 half-life (Fig. 6B–D). Furthermore, analysis of published ChIP-seq data[49–51] for SOX2 and ISL1 showed both SOX2 and pSMAD1 bind enhancers near ISL1 as well as the SOX2 promoter, consistent with our model (Supplementary Fig. 5L) and previous work showing direct regulation of SOX2 by BMP signaling[52–54]. More generally, our results illustrate the signaling memory of the system can be much longer than the lifetime of proteins involved.

To test the role of SOX2 directly we then created a doxycycline-inducible SOX2 cell line (Supplementary Fig. 5M, N). We found that doxycycline-induced SOX2 overexpression for all 42 h of differentiation prevented upregulation of amnion markers, suggesting SOX2 represses these genes (Supplementary Fig. 5O, P). However, NANOG expression was also lost, consistent with the known requirement for the right stoichiometry between pluripotency genes to maintain pluripotency[55,56]. To stay within the range of SOX2 levels where pluripotency is possible, we then treated cells with doxycycline for 12 h after 12 h of BMP treatment, when endogenous SOX2 levels have already decreased significantly. We found that this significantly decreased differentiation to ISL1+ amnion but maintained pluripotency in the ISL1- cells (Fig. 6E–H). We repeated this experiment in micropatterned colonies and found that SOX2 overexpression reduced ISL1 but not GATA3, consistent with our model wherein only delayed differentiation genes are controlled by the integral threshold mediated by SOX2 (Fig. 6I–K, Supplementary Fig. 5Q).

## Discussion

We showed that the time-integral of BMP signaling determines amnion-like fate in human pluripotent stem cells. Importantly, this challenges the idea of level thresholds controlling differentiation since the same integrated signaling level can be reached by short high signaling and long low signaling. These findings therefore potentially have broad repercussions for our understanding of developmental patterning, while providing specific insight into early human cell fate decisions and heterogeneous stem cell differentiation in vitro. We identified 'integrator genes', whose levels reflect the time integral of BMP and provide evidence that SOX2 mechanistically implements the time-integration of BMP signaling. While the full GRN interpreting BMP is likely more complex and remains to be elucidated, the linear relationship between integrator gene expression and signaling is not generically expected and is in some sense the opposite of sharp level thresholds. Therefore, their identification provides additional support to the integral model.

To investigate how signaling is interpreted by cells we leveraged spontaneous signaling heterogeneity. Supplementary Fig. 2D, E and previous work suggest signaling heterogeneity is due to receptor localization and inhibitor secretion combined with local differences in cell density and confluence[28]. However, our conclusion that integrated signaling controls differentiation does not depend on how signaling heterogeneity arises.

In interpreting our data, we made several idealizations. First, the level of SOX2 and similar integrators cannot perfectly reflect the signaling integral due to finite turnover as well as cross- and auto-regulation. Our model (Fig. 6D) showed that this is a good approximation (Fig. 6B–D), although larger deviations from an exact integral threshold are predicted for low levels combined with longer durations than we experimentally tested. Second, we used the term fate for ISL1+ amnion-like cells versus SOX2+ pluripotent cells, but as this is the first step of differentiation from pluripotency in a long series of developmental events, we do not expect the differentiation markers to truly mark a stable cell fate. Rather, our amnion-like cells may represent an intermediate state towards further differentiation. Nevertheless, we showed commitment to differentiation, as the pluripotency gene SOX2 does not recover even after removal of BMP in cells that pass the threshold for differentiation (Fig. 6).

There are several potential advantages to integrating cell signaling over time. Integration reduces noise and is insensitive to brief signaling perturbations in the same way as averaging. It also allows for flexible tuning of fate patterns, since an integral threshold is equivalent to a level threshold that can be tuned by changing the duration, e.g. with a delayed negative feedback loop, which may be more straightforward than scaling the morphogen gradient itself. This also implies that if the duration of signaling is the same for all cells, a signaling gradient interpreted by integration over time would produce cell fate patterning consistent with the classic French flag model despite the absence of absolute thresholds in signaling level.

Given these advantages, temporal integration of cell signals might be expected throughout development. Indeed, there is evidence for integration of ERK signaling in mammary epithelial cells[57], and Nodal signaling in zebrafish[58]. In many other contexts where a strong dependence on signal duration was found, this could be a dependence on integrated signaling, depending on whether duration thresholds change with signaling level. For example, our finding that BMP dose primarily affects the average duration of signaling (Fig. 3) is reminiscent of SHH signaling in the neural tube, where concentration increases increase the duration but not the level of signaling[18,20]. Strikingly, BMP signal duration is also important in neural tube patterning, but it remains unclear whether dorsal interneuron identity depends on the time-integral of BMP signaling or on level, duration, and other signaling features separately[17]. In addition to the time-integral of signaling possibly controlling fate more generally, the potential mechanism of BMP integration by SOX2 may also extend to other cell fate decisions, since several fate decisions controlled by BMP involve suppression of SOX2 including neural versus non-neural ectoderm[59–61] and foregut versus hindgut[62]. An important related question is whether our inability to obtain steady response at different

### A simple mathematical model explains differentiation driven by SOX2 dynamics

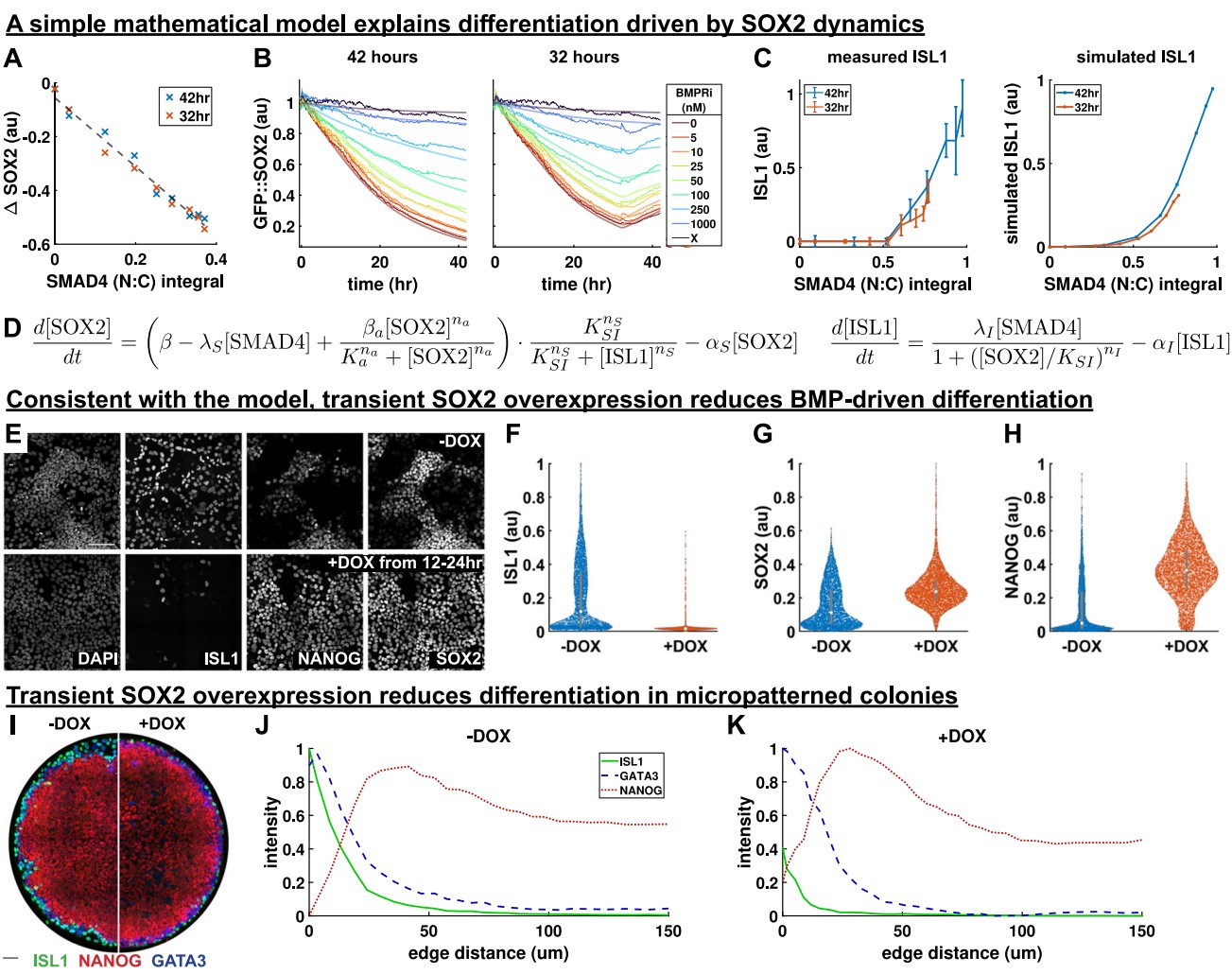

**D**
$$\frac{d[\text{SOX2}]}{dt} = \left( \beta - \lambda_S[\text{SMAD4}] + \frac{\beta_a[\text{SOX2}]^{n_a}}{K_a^{n_a} + [\text{SOX2}]^{n_a}} \right) \cdot \frac{K_{SI}^{n_S}}{K_{SI}^{n_S} + [\text{ISL1}]^{n_S}} - \alpha_S[\text{SOX2}] \qquad \frac{d[\text{ISL1}]}{dt} = \frac{\lambda_I[\text{SMAD4}]}{1 + ([\text{SOX2}]/K_{SI})^{n_I}} - \alpha_I[\text{ISL1}]$$

### Consistent with the model, transient SOX2 overexpression reduces BMP-driven differentiation

### Transient SOX2 overexpression reduces differentiation in micropatterned colonies

**Fig. 6 | SOX2 perturbation reduces amnion-like differentiation in agreement with a mathematical model. A** Plot of the change in each GFP::SOX2 curve in Supplementary Fig. 5H over the first 16 hours of differentiation against SMAD4-signaling integral in that time. SMAD4 integral is determined in each condition based on data in Fig. 4. A dashed line indicates a linear fit of the data. **B** Measured (thick, solid lines) and simulated (thin, semi-transparent lines) GFP::SOX2 dynamics over the course of 42 hours of differentiation with indicated treatments applied for 42 (left) or 32 (right) hours. **C** Measured (left) and simulated (right) ISL1 level as a function of SMAD4 integral for the conditions in **B**. Data presented as mean +/- SD across $N = 4$ images. **D** Equations used to model the regulation of SOX2 and ISL1 by BMP-SMAD4 signaling, and their mutual inhibition. **E–H** Example IF data (**E**) and

quantification showing ISL1 (**F**), SOX2 (**G**), and NANOG (**H**) expression after 42 hours of differentiation in standard culture with 50 ng/mL BMP4 + WNTi, +/- doxycycline from 12 to 24 hours. Violin plots show expression distributions over $n = 2867$ cells (-DOX) and $n = 3081$ cells (+DOX) in one of two independent experiments. The box plot within the violin shows the 25%, 50% (median), and 75% quartiles, with whiskers to the maximum and minimum values in the data. **I–K** Example IF data (**I**) and radial expression profiles (**J, K**) in micropatterned colonies after 42 h of differentiation with 50 ng/mL BMP4 + WNTi, without (**J**) or with (**K**) the addition of doxycycline from 12 to 24 hours. Scale bars 50um. Source data are provided in a Source Data file.

levels by varying exogenous BMP concentration is typical or depends on developmental stage and the combination of BMP ligands and receptors[63,64].

Several papers previously considered the role of BMP signaling in hPSC differentiation. Gunne-Braden et al.[45] claimed that GATA3 mediates fast, irreversible commitment to differentiation after less than 1 h of BMP exposure. In sharp contrast, our work shows that a substantial duration (over 24 h) of BMP signaling is required for both differentiation and high GATA3 expression even at maximal signaling levels. Moreover, Supplementary Fig. 4V–X and Fig. 5 show a gradual relationship between GATA3 expression and the duration of BMP signaling that supports our integral model and is inconsistent with switch-like behavior. Consistent with our data, a large body of literature supports the conclusion that BMP inhibition at any time during differentiation has a clear impact on cell fate[27,28,34,65–67].

Tewary et al.[32] also studied how BMP concentration and duration affect micropatterned hPSC colonies. However, they only

mathematically modeled BMP gradient formation and did not quantitatively investigate the relationship between signaling and fate. Qualitatively, they proposed final pSMAD1 levels determine fate boundaries, but this seems inconsistent with their finding that marker genes are expressed earlier at higher doses of BMP. BMP signaling in micropatterned colonies does not increase over time (Fig. 1) so cells above a pSMAD1 level threshold for a given marker would be expected to be always above it and thus show similar expression dynamics. In contrast, the integral model predicts earlier expression at higher signaling levels. Their data therefore appear consistent with our integral model.

Nemashkalo et al.[34] showed duration rather than the initial BMP response correlates with fate at the population level and proposed a duration threshold. Although they performed single-cell tracking, this was for a small number of cells in a single condition and not for the full duration of differentiation. Consequently, they were unable to quantitively relate signaling to fate and did not demonstrate a fixed

duration threshold. They also did not study large micropatterned colonies that model embryonic patterning. In contrast, here we showed there is no duration threshold but rather an integral threshold and we provide a downstream mechanism that quantitatively explains fate from signaling in both standard culture and micropatterned colonies.

Finally, concurrently with and complementary to this work, Camacho-Aguilar et al.[31] explored the effect of concentration versus duration of BMP treatment on cell fate in a different context where BMP acts combinatorially with downstream Wnt to control a decision between primitive streak-like and amnion-like fate. In contrast, we were able to determine which features of BMP signaling history control gene expression only by inhibiting Wnt. However, their results are consistent with ours in showing that different concentrations of BMP primarily affect the duration of response.

Future work will have to address the molecular mechanism of BMP integration in more detail. For example, it is not clear how the rate of SOX2 decrease is controlled by BMP signaling and how SOX2 represses late differentiation genes, including ISL1. Perhaps most importantly, we do not understand why there are early differentiation genes, including TFAP2C and GATA3, which also meet all the requirements to be integrator genes in addition to late differentiation genes such as ISL1 and HAND1 whose expression appears to mark a commitment to differentiation. One possibility is that these genes act as integrators complementary to SOX2, e.g., our data are consistent with the possibility that ISL1 upregulation requires both SOX2 down-regulation and GATA3 upregulation. Either way it seems likely SOX2 is embedded in a larger gene regulatory network than we were able to explore.

There has been debate about whether heterogeneous differentiation in human pluripotent stem cells is primarily due to a het-erogeneous initial state or heterogeneous signaling response[68] and several papers found initial levels of pluripotency markers to be pre-dictive of differentiation[36,37] although the simplest models for tissue patterning assume a fixed relationship between signaling and fate. Intuitively it is clear that both should matter and that it depends on the specific context, which dominates. However, our findings unify these contrasting results in the literature more concretely by providing a direct connection between the initial levels of the pluripotency factor SOX2, which sets the threshold for integrated BMP signaling, allowing calculation of the relative contributions of signaling heterogeneity and initial heterogeneity in SOX2.

Our tracking approach in standard culture is easily generalized to widely used differentiation protocols and future work will investigate whether the same mechanism for BMP interpretation is reused at different developmental stages. Equally important is investigating at the single-cell level how combinatorial signaling histories are interpreted, for example, in the human germline, where the relative timing and duration and BMP and Nodal signaling appear to play a key role in determining fate.

## Methods
### Cell lines
We used the embryonic stem cell lines ESI017 (XX; ESI BIO) and RUES2 (XX; gift of Ali Brivanlou, Rockefeller). Genetically modified variants of these two cell lines were RUES2 GFP::SMAD4 (Nemashkalo et al., 2017; gift of Ali Brivanlou), RUES2 RFP::SMAD1 (Yoney et al., 2018; gift of Ali Brivanlou), and ESI017 tetO-SOX2 (this paper). We additionally used the genetically modified induced pluripotent stem cell line WTC11 GFP::SOX2 (XY; Allen Institute, identifier AICS-74; RFP nuclear marker added in this work).

### Cell culture and differentiation
Human pluripotent stem cells were cultured in the pluripotency-maintenance media mTeSR1 (StemCell Technologies) on Cultrex (R&D

Systems)-coated 35 mm tissue culture plates. During routine main-tenance, cells were passaged every 2-4 days, either in whole colonies with L7 [69], or in single-cell suspension with Accutase. Seeding for differentiation experiments was done in single-cell suspension after dis-sociation with Accutase. When passaging in single-cell suspension with Accutase, cells were kept for 24 hours in ROCK inhibitor (RI) after passaging. RUES2 GFP::SMAD4 cells were selected with 24 hours of blasticidin treatment at every passage. Blasticidin was removed 24 hours after passaging, and was always excluded during differ-entiation experiments.

For differentiation experiments in standard (not micropatterned) culture, cells were passaged and seeded into μ-Slide 18-well plates from Ibidi (with the exception of bulk RNA seq experiments, which used 24-well Ibidi plates) 16–20 hours before treatment with BMP. When performing live-cell imaging, media was changed from standard mTeSR to mTeSR without phenol red 2.5 hours before BMP4 treat-ment. For experiments in which cells were not maintained in RI over the course of differentiation, RI was removed 2.5 hours before treat-ment time. For experiments in Fig. 3, cells were sparsely labeled to facilitate automated single-cell tracking. For sparse labeling, 10-20% GFP::SMAD4 RUES2 cells were mixed thoroughly with 80-90% wild-type RUES2 cells and seeded at the desired density.

To differentiate cells in micropatterned colonies, we followed the procedure described in Jo et al.[30], adapted from the protocol in Deglincerti et al.[70]. Briefly, cells were dissociated with Accutase, resuspended in a single-cell suspension, and seeded at 470k cells/cm$^2$ onto laminin-coated micropatterns in mTeSR with RI. Colonies were washed 2x with PBS$-/-$ 45 minutes after seeding to clear away those binding non-specifically in the well outside of micropatterned colo-nies. Two hours after seeding, RI was removed and BMP4 treatment added. Micropatterning experiments were performed in 18-well Ibidi slides prepared according to the protocol in Azioune et al., 2009[71]. All micropatterned colonies had a diameter of 700 μm. Signaling reagents and treatment concentrations are listed in Supplementary Table 1. Cells were routinely tested for mycoplasma contamination and nega-tive results were recorded.

### Control of signaling level and duration
In experiments in which we controlled the level and duration of SMAD4 signaling at the population level, we used a modified experi-mental protocol. Cells were seeded in Ibidi μ-Slide 18-well plates at a low density of $1.5 \times 10^4$ cells/cm$^2$ 16–20 hours before treatment. To ensure uniformly high signaling, in addition to sparse seeding density, cells were maintained in RI over the course of differentiation and treated with a high BMP4 dose of 100 ng/mL. The signaling level was tuned by varying the concentration of BMPRi (LDN193189) added concurrently with BMP4, and duration was controlled by removing BMP4 and adding a saturating dose of 1000 nM BMPRi. Wnt ligand secretion inhibitor WNTi (IWP2) was included over the course of dif-ferentiation in all such experiments. A similar approach was used to generate the pulses of signaling in Fig. 4I, with inhibitor washed out and replaced with 100 ng/mL BMP4 to upregulate signaling again after inhibition. However, to inhibit BMP4 signaling we used 250 ng/mL NOGGIN instead of LDN193189, as we found that signaling was not increased back to maximal level after washing and restoring a high dose BMP4 without LDN, suggesting that it is difficult to wash out LDN completely.

### Immunofluorescence staining
Samples were washed twice in PBS without calcium and magnesium (PBS$^{-/-}$), fixed with 4% paraformaldehyde for 20 minutes at room temperature (RT), and then washed with PBS$^{-/-}$ two more times. They were then incubated in a permeabilization buffer (0.1% Triton X-100 in 1X PBS$^{-/-}$) for 10 minutes at RT and rinsed twice more with PBS$^{-/-}$. Cell lines expressing fluorescent proteins were photobleached according

to Lin et al., 2015 by incubating for 1 hour at room temperature in a bleaching buffer (3% $H_2O_2$, 20 mM HCL diluted in $PBS^{-/-}$) under an incandescent lamp, followed by two washes with $PBS^{-/-}$. After permeabilization and optional bleaching, blocking was done with a blocking buffer (3% donkey serum + 0.1% Triton X-100 diluted in 1X $PBS^{-/-}$) for 30 minutes at RT. Following blocking, samples were incubated overnight at 4 °C in a solution of primary antibodies diluted in blocking buffer (antibodies and dilutions are listed in Supplementary Table 2). Following primary antibody incubation, samples were washed 3x with PBST (0.1% Tween 20 in 1X $PBS^{-/-}$), with 20 minutes incubation at RT between each wash. Samples were incubated in a solution of secondary antibodies diluted in blocking buffer (antibodies and dilutions described in Supplementary Table 3) and DAPI (1 µg/mL; ThermoFisher Scientific) for 30 minutes at RT in the dark. After incubation, two fast PBST washes were performed, followed by two PBST washes with 20 minutes incubation at RT between each. Cells were stored in $1xPBS^{-/-}$ with 0.01% sodium azide, and were transferred to imaging buffer (700mM N-Acetyl-Cysteine in ddH2O, with pH adjusted to 7.4; Gut et al., 2018[40]) immediately before imaging to prevent photo-crosslinking of antibodies.

### Generation of the ESI017 tetO-SOX2 cell line

An enhanced piggyBac Puromycin selectable and DOX inducible vector was digested with EcoRI-NotI and ligated with PCR-amplified human SOX2 from the FUW-tetO-hSOX2 plasmid (Addgene#20724). Transfection into hPSCs was done using Lipofectamine Stem Reagent per the manufacturer's instructions. Transfected cells were selected with 1ug/ml Puromycin for 1 to 2 weeks until only single colony clones remained.

### Repeat staining

To perform iterative immunofluorescence, we adapted the protocol described in Gut et al.[40] for elution and sample re-staining as described in Freeburne et al.[72]. After the previous iteration of immunofluorescence imaging, the sample was washed three times with $PBS^{-/-}$ and three rounds of antibody elution were performed; in each round the sample was incubated for ten minutes at room temperature in an elution buffer (0.5M L-Glycine, 3 M urea, 3 M guanidine hydrochloride, and 70 mM TCEP, diluted in ddH2O, pH adjusted to 2.5) while being shaken at 50 rotations per minute (RPM) on a tabletop orbital shaker. The sample was washed three more times with $PBS^{-/-}$, and blocked on the orbital shaker in a blocking buffer for 30 minutes at room temperature. Primary antibody dilutions and incubation were done as for initial IF staining. Secondary antibody staining was performed as in initial staining, with the incubation in the solution of secondary antibodies extended from 30 min to an hour, and done on an orbital shaker at 50 RPM. Storage and imaging buffers are as for initial IF staining.

### Imaging

Imaging was performed with an Andor Dragonfly/Leica DMI8 spinning disk confocal microscope with a ×40, NA 1.1 water objective and a x20 air objective using Andor Fusion software version 2.3.0.31, as well as a Nikon/Yokogawa spinning disk confocal microscope with a x20 air objective using NIS Elements AR software version 5.41.02. Live-cell imaging was performed with controlled temperature (37°), $CO_2$ concentration (5%), and humidity (>60%). Experiments for which single-cell tracking was performed (Fig. 3) were performed using the x40 water objective. Other experiments were generally performed with a x20 air objective. Live-cell experiments in disordered culture were imaged every 10 minutes, using a z stack with 4 slices spaced 3 to 3.33 microns apart. Live-cell imaging of micropatterned colonies was done with a time interval of 20 minutes using a z stack with 4 slices spaced 4 to 5 microns apart. Media and treatment changes were performed in the time between imaging intervals without removing the sample from the microscope stage. For experiments in which both live and fixed-cell image data was quantified for the same cells, the same microscope and objective was always used; upon the conclusion of live-cell imaging, the sample was immediately taken for fixation to minimize the movement of cells between the conclusion of live-cell imaging and fixation and facilitate matching of live to fixed nuclei. PFA was added to the sample within 10 minutes of the conclusion of live imaging.

### Image analysis

In immunofluorescence data, nuclei were segmented based on DAPI staining using both Cellpose[73] (v1) and the pixel classification workflow in Ilastik[74] (v1.3.3post2). Ilastik and Cellpose masks were merged into a single segmentation as previously described[30]. After consolidating these segmentations, the object classification workflow in Ilastik was used to identify and discard missegmented (junk) objects. Cells in disordered culture form a monolayer and their nuclei were segmented based on the maximal intensity projection (MIP) of the z stack of DAPI images. For micropatterned colonies in which cells may be layered two or three cells deep, we segmented nuclei in each z slice and merged nuclear masks across z slices into a single 3D segmentation as previously described[30].

In live imaging montages of micropatterned colonies, nuclei were segmented in the same way as in immunofluorescence data. In live cell images in sparser disordered culture, a nuclear segmentation pipeline optimized for single-cell tracking using only Ilastik was used. To facilitate single cell tracking after pixel classification, an additional step of Ilastik object classification marked segmented objects as interphase, metaphase (chromosomes aligned along the metaphase plate, immediately prior to splitting), other dividing (prophase - chromatin is visibly condensed but not aligned; anaphase - sister chromatids are moving apart but may not yet be segmented as two separate objects). For immunofluorescence data, an additional class was included to discard missegmented objects. Finally, a custom algorithm for approximate convex decomposition[30] was applied to interphase-labeled foreground objects in the nuclear segmentation to split overlapping or touching nuclei into distinct masks.

Downstream quantification was carried out with a custom image-processing pipeline written in MATLAB (versions 2019b – 2023a). Expression levels were calculated as mean intensity in each channel within the nuclear mask. Both for segmentations of the MIP and 3D segmentations, the intensity was quantified for each nucleus in the z slice in which that nucleus was most in-focus, determined based on the intensity profile of the nuclear marker (DAPI or H2B) in z.

In live-cell images of GFP::SMAD4 or RFP:SMAD1, we additionally used Ilastik pixel classification to segment cell bodies as foreground, and used the inverse of this mask to detect the image background. To determine cytoplasmic intensity for each cell, a watershed operation was performed with nuclear masks imposed as minima for the watershed. For each nucleus, a cytoplasmic mask was constructed as an offset annulus about the nucleus, intersected with both the watershed basin corresponding to that nucleus and foreground mask of SMAD4 or SMAD1. Values in the cytoplasm were calculated as mean intensity in the cytoplasmic mask in the same z slice in which nuclear intensity was computed. SMAD4 nuclear to cytoplasmic ratio was taken as background-subtracted nuclear intensity divided by background-subtracted cytoplasmic intensity. For RFP::SMAD1, the cytoplasmic values were very close to background, so we directly computed nuclear to cytoplasmic ratio without background subtraction to reduce sensitivity to noise.

For the quantification of multiple rounds of immunofluorescence staining and imaging, phase correlation-based image registration was used to find a rigid shift aligning consecutive rounds of imaging to the first round. For cells in disordered culture, the same segmentation was used from the first round of imaging to quantify expression in subsequent rounds after alignment. For micropatterned colonies, a 3D

segmentation was generated for each round of imaging separately, and individual cells were linked between rounds based on aligned x and y and normalized z centroid positions of each cell, using the algorithm for matching live to fixed cells described in the single-cell tracking supplement (Supplementary Note 2).

In micropatterned colonies of hPSCs, we performed analysis based on edge distance by subdividing the colony into 30 bins with equal numbers of cells in each bin. In each bin, all cells within that bin are within a similar distance from the colony edge. The average expression or signaling value for each bin was taken as the median among the cells in that bin.

For the RFP::SMAD1 and GFP::SMAD4 cells mixed in the same colonies in Supplementary Fig. 1T-W, the same excitation captured images of GFP::SMAD4 and the H2B::mCitrine nuclear marker of the RFP::SMAD1 cells. Likewise, the same excitation captured RFP::SMAD1 and the H2B::RFP nuclear marker of the GFP::SMAD4 cells. Because fluorescence intensity of the tagged H2B proteins was much higher than that of the SMAD proteins, we were able to use Ilastik pixel classification to classify pixels in each channel as nuclei, SMAD, background, or junk. Quantification of nuclear to cytoplasmic intensity for each signaling protein was then performed as described above, with intensities in a small radius around nuclei of SMAD4 cells excluded from the SMAD1 quantification and vice-versa. The quantification of cells in the two lines was consolidated, with a label added for cell type for the downstream analysis in Supplementary Fig. 1U-W. To make the visualization in Supplementary Fig. 1T, nuclear segmentation masks were used to computationally separate the much brighter H2B::RFP and H2B::mCitrine signal from RFP::SMAD1 and GFP::SMAD4 and combine them into a contrast-adjusted false-color image.

### Single-cell tracking

Fully automated single-cell tracking was performed with a custom algorithm written in MATLAB described in detail in Supplementary Note 2, modified from Jaqaman et al. [38] and similar to the implementation used in Trackmate[75]. Most importantly, to better handle cell division we applied machine learning[74] to label nuclei as dividing based on morphological and image intensity information, and adjusted the linking function to handle nuclei during and after cell division (Supplementary Fig. 2). For each single-cell tracking experiment, a subset of cell tracks were manually validated and results for the larger dataset were corroborated with the subset of validated tracks. Live cells were matched to fixed cells using the same algorithm used for tracking live cells as described in Supplementary Note 2.

### Analysis of signaling histories

Analysis of single-cell signaling histories was carried out in MATLAB and Python. In Fig. 1, clustering of signaling histories was done using soft k-means with k = 3 in MATLAB with fcm. To compare the fate pattern to the signaling cluster pattern, we discretized the profile of fate markers, assigning the most prevalent fate at each position, and then averaged this over multiple colonies as a way to visualize (minimal) variation between colonies (Fig. 1J, Supplementary Fig. 1G). The prediction of the fate boundary from signaling was initially less accurate with Wnt inhibitor than without (Supplementary Fig. 1O–R), but we found that this could be attributed to the fact that there is no objective way to assign the elbow fate, and the clustering algorithm produced the wrong assignment. Manually changing the cluster assignment of the elbow led to closer agreement with the fate pattern (Supplementary Fig. 1S), and this is what we showed in main Fig. 1LM.

In Fig. 3, signaling features were fit in MATLAB using lsqnonlin. Single-cell histories were denoised in Python with MAGIC (Markov Affinity-based Graph Imputation of Cells) using three nearest neighbors (knn = 3) and the diffusion operator to third power (t = 3).

Clustering by fate was first performed by fitting a two-component Gaussian mixture model to the seven-dimensional immunofluorescence data. We determined which markers best separate the clusters by calculating cluster separation as the difference in the means over the sum of the standard deviations for a specific marker (Supplementary Fig. 3H, I). As an alternative approach, we also processed our seven-dimensional immunofluorescence data in the same way as single-cell RNA-sequencing data, clustered it with the Leiden algorithm and calculated differential expression between the clusters (Supplementary Fig. 3J, K). Both approaches yielded ISL1 and NANOG as top genes.

To determine the relationship between signaling and fate, we had the option of denoising both history and fate based on the cells with most similar fate marker expression, the cells with the most similar signaling histories, or some linear combination of the two. Therefore, we compared the two extremes and to ensure we were not simply creating artificial correlations included a control where signaling histories were randomly assigned to marker expression before denoising, which reassuringly did not yield any correlation (Supplementary Fig. 3M, bottom). We found that denoising based on fate yielded higher correlation between signaling and fate and better preserved the bimodal distribution of fate markers (Supplementary Fig. 3M). Moreover, this approach is conceptually appealing because it directly extends the averaging over all cells with two discrete fates in Fig. 3H to essentially more fine-grained averaging of histories between small numbers of cells with most similar fate marker expression. We therefore applied fate-based denoising for combined analysis of signaling and fate.

To test how much information each feature contains about fate we determined the accuracy (% true positives + true negatives) of a Bayesian classifier, which is formally optimal[76] and determines the most probable fate given the value of a signaling feature from the conditional probability P(fate|feature). Because of the monotonic relationship between fate and features this came down to determining an optimal threshold in the signaling feature. The four quadrants made by the fate threshold and the signaling threshold then provide the confusion matrix of the resulting binary classifier, with amnion-like cells (log(ISL1/NANOG) > 0) above/below the signaling feature threshold corresponding to true/false positive predictions, and pluripotent cells (log(ISL1/NANOG) < 0) above/below the signaling feature threshold corresponding to false/true negatives, respectively. From this one can then formally calculate the information contained about fate in the signaling features as a single number called the decoder-based mutual information[77], which has the nice property that it is zero for pure chance, whereas the total accuracy (true positives + true negatives) is 50% for pure chance in this binary classification, but for simplicity we chose to present the accuracy.

We compared the Bayesian fate prediction based on integrated signaling to fate prediction based on the full signaling history using generic classifiers. Specifically, we trained various neural networks implemented using PyTorch and Support Vector Machines using scikit-learn. Performance was evaluated using 5-fold cross-validation. The neural network results shown in Fig. 3P correspond to a network without hidden layers and sigmoid activation function. We found that addition of a 20-neuron hidden layer led to similar performance. The support vector machine result in Fig. 3P used a quadratic kernel.

### RNA sequencing and analysis

For RNA sequencing of the time series after BMP treatment, total RNA extraction was performed with the Invitrogen RNAqueous micro kit according to the manufacturer's instructions. Cells were collected and lysed with the provided lysis buffer at specified times and the lysate was frozen and stored at −80℃. Whole RNA was prepared and DNase-

treated for all samples at the same time, per kit instructions. The University of Michigan Advanced Genomics Core performed library preparation for mRNAs with ribosomal RNA depletion, and sequencing was performed in an Illumina NovaSeq S4 Flowcell with a sequencing depth of 57 M reads per sample.

For the dose-response, total RNA extraction was performed with the QIAGEN RNeasy micro kit according to the manufacturer's instructions. Five hours after BMP4 treatment cells were lysed with lysis buffer RLT and lysate was collected. Whole RNA was prepared and DNase-treated according to the kit instructions. The University of Michigan Advanced Genomics Core performed library preparation for mRNAs with polyA selection, and sequencing was performed in an Illumina NovaSeq S4 Flowcell with a sequencing depth of 33 M reads per sample.

Reads from FASTQ files were trimmed using Cutadapt v2.3 (Martin, 2011) and mapped to the reference genome GRCh38 (ENSEMBL), using STAR v2.7.8a (Dobin et al., 2013). Count estimates were generated with RSEM v1.3.3 (Li and Dewey, 2011). Alignment options followed ENCODE standards for RNA-seq.

For analysis of time-series sequencing data, low-expressed genes were defined as those with less than 2.5 counts per million averaged over all conditions and filtered out. We further filtered out those showing relatively little change by keeping only genes with an absolute cumulative log2 fold change greater than -1.55 (one standard deviation). After filtering, each gene was normalized to its maximum value in the time series. Agglomerative hierarchical clustering was performed in MATLAB after normalization using euclidean distance and Ward's linkage.

### ChIP-seq analysis
Chip-seq analysis was performed in Integrative Genomics Viewer[78]. GEO reference numbers GSE187636[49], GSE61475[51], GSE17312[50].

### Mathematical modeling
Dynamics of SOX2 and ISL1 expression were simulated with a nonlinear system of two first-order ordinary differential equations. Idealized SMAD4 dynamics with levels inferred from live imaging experiments were used as input to the model, and numerical simulations were performed in MATLAB. Rationale for the model construction and details of fitting to expression data are described in Supplementary Note 1.

### Fluorescence recovery after photobleaching (FRAP)
We used FRAP to measure the half-life of GFP::SOX2. Photobleaching of GFP::SOX2 was carried out using a Zeiss LSM800 confocal microscope (Zeiss, Germany). A square region of interest was photobleached by scanning with the 488 nm laser at full (100%) power, requiring 10 seconds to photobleach most of the GFP::SOX2 fluorescence. Image acquisition before and after the photobleaching was carried out using the Andor Dragonfly / Leica DMI8 spinning disk confocal microscope. Photobleaching and transferring the sample between microscopes took 10-15 minutes. We quantified mean GFP::SOX2 fluorescence intensity in cells inside the bleached region relative to mean intensity in cells in a region of interest far outside of the bleached region. To determine protein half-life from the measured dynamics, we fit an exponential function of the form $A_0 + A(1 - e^{-\lambda t})$ and determined half-life as $\ln(2)/\lambda$ where ln is the natural log. Because FRAP reduces only fluorescence of SOX2 rather than its total level, recovery from photobleaching results from turnover in equilibrium and follows a simple exponential, unaffected by the positive autoregulation that we included in our model.

### Statistics and reproducibility
All experiments were performed at least twice with the exception of the RNA sequencing in Fig. 5. All attempts at replication were successful. The results shown in Fig. 1C were replicated in more than 10

independent experiments. The results in Fig. 1D were replicated in 3 independent experiments. Figure 2 results were replicated in 5 independent experiments. Similar results to those shown in Fig. 5D and Fig. 6E, I were obtained in 2 independent experiments each.

### Reporting summary
Further information on research design is available in the Nature Portfolio Reporting Summary linked to this article.

## Data availability
RNA-sequencing data generated in this study have been deposited in GEO under accession number GSE229675. Minimal processed data and code to reproduce key results is on GitHub, linked in Code Availability. Raw image data are available upon request but are too large to practically host in a public repository. Source data are provided with this paper.

## Code availability
All code for data analysis and model simulations is available on https://github.com/idse/BMPintegral and on Zenodo https://doi.org/10.5281/zenodo.10076773[79].

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

## Acknowledgements

We thank Mara Duncan, Ben Allen, Doug Engel, and Aryeh Warmflash for discussions and feedback on the manuscript. This work was supported by the National Institute of General Medical Sciences (NIGMS R35GM138346), NSF RECODE (2033654), the Branco Weiss Fellowship – Society in Science, and the University of Michigan. EF was partially supported by the NIH Cellular Biotechnology Training Program (T32GM145304) and KJ was partially supported by the Michigan Pioneer Postdoctoral Fellowship and the NIH F32 Ruth L. Kirschstein Postdoctoral National Research Service Award (5F32HD108980-02).

## Author contributions

The study was conceptualized by I.H. and S.T. Experimental methodology was developed by S.T., K.J., B.C. Experiments and data collection were carried out by S.T., G.P., B.C., Z.Y.L., L.A.Y., E.F., and H.K. Data analysis and interpretation were performed by S.T., Z.Y.L., C.J., and I.H. Software for data processing and analysis was developed by S.T., Z.Y.L., and I.H. Writing – original draft, and writing – review and editing were done by I.H. and S.T. Funding was acquired by I.H. All authors reviewed and commented on the manuscript.

## Competing interests

The authors declare no competing interests.

## Ethics

Our research complies with all relevant ethical regulations. Approval for work with human embryonic stem cells and human induced pluripotent stem cells was granted by the Human Pluripotent Stem Cell Research Oversight (HPSCRO) Committee at the University of Michigan.
