## [Peer Review File · Nature Communications]

Time-integrated BMP signaling determines fate in a stem cell model for early human developmentREVIEWER COMMENTS

Reviewer #1 (Remarks to the Author):

In this manuscript, Teague et al. investigate the relationship between single cell signaling histories and cell fate, focusing on the decision between amnion-like and pluripotent cells controlled by BMP in the absence of WNT. The authors developed an imaging pipeline with automated tracking and impressive multiplexed immunofluorescence stainings, resulting in a large dataset of signaling histories linked to cell fate. They conducted experiments with different signaling levels and durations, and observed that the duration threshold for differentiation changed depending on the signaling level, but the integrated signaling remained the same. The authors then explored possible mechanisms by which cells might integrate BMP signaling and identified SOX2 as a potential integrator. SOX2 levels were found to reflect integrated SMAD4 signaling in an inverse relationship, and a simple model incorporating SOX2 production, degradation, and negative regulation by ISL1 showed good agreement with the experimental data. Finally, using a doxycycline-inducible SOX2 cell line, the authors demonstrated that maintaining SOX2 levels by brief doxycycline treatment prevented upregulation of amnion markers. Overall, the work is relevant for the fields of development and stem cell biology, but the finding that both signaling level and duration are important for morphogen interpretation is not new and has been demonstrated in multiple systems. Several predictions in the current manuscript are based on correlations rather than direct tests, and additional evidence is needed to substantiate the claims and resolve conflicts with other studies as detailed below.

MAJOR POINTS

1) Readout of BMP signaling

- a) There are concerns regarding the choice of SMAD4 as a readout for BMP signaling. The SMAD1 and SMAD4 graphs in Fig. 1E and Fig. S1D indicate that SMAD4 is not a good readout for BMP activity since SMAD1 is expressed uniformly until about 17 h (Fig. S1D,H), whereas SMAD4 is only expressed uniformly until about 11 h (Fig. 1E).
- b) The role of Nodal in these experiments should be tested conclusively with a Nodal inhibitor, particularly given the findings by Guglielmi et al. 2021 regarding the role of SMAD4 to mediate Nodal and BMP signaling.
- c) Given these concerns, it will be important to repeat the key findings of the current work with the SMAD1-RFP cell line to bolster the conclusions.
- d) The claim “In other words, different concentrations of BMP lead to different signaling durations but not different signaling levels” needs to be corroborated by checking phosphorylated SMAD1/5/9.
- e) If projected on a different color range, the tails in Fig. 1E and Fig. S1D,H at $t > 40$ h might look similar, arguing for a more complicated signaling landscape. Please clarify by ruling out potential normalization issues, and explore different color ranges/thresholds to address this concern.

2) Integration model

- a) The paper claims that BMP signaling is integrated over very long times scales of more than 40 hours.

This strong claim needs more conclusive tests. If the signal is truly integrated, recurring hour-long pulses of inhibition and strong activation should give the same outcome in terms of amnion differentiation as the an integral of the same magnitude that provides continuous activation at a lower level. This additional test of the integration model is also important to support the authors' statement that "Integration reduces noise and is insensitive to brief signaling perturbations in the same way as averaging".

b) The paper overlaps to a good degree with the preprint from Camacho-Aguilar et al. 2023 who also show integration in time and state that "SOX2 decay dynamics show a very high correlation with the integral of SMAD4 in time". However, the conclusion of the paper "We showed a BMP-induced morphogen effect in the duration but not the concentration of signaling, indicating that duration and concentration of BMP signal are not interchangeable in this context" is different from the conclusions of the current work. It will therefore be necessary to more conclusively test the integration model with the pulses described above to resolve this conflict.

c) The prediction that overexpression of SOX2 reduces differentiation to amnion-like fate in response to BMP is not a strong one and appears to be compatible with multiple morphogen interpretation models, not exclusively the integration model. If the authors disagree, they should more explicitly describe in the text why the outcome of the experiment is incompatible with other morphogen interpretation models.

d) The authors selected SOX2 as an integrator candidate, but the evidence is not strong. Additional data would strengthen this hypothesis. For example, what is the experimental evidence that SOX2 directly represses ISL1, and does ISL1 repress SOX2? Is SOX2 a direct target of pSMAD, and does pSMAD bind in SOX2 promoter/enhancer regions?

3) Methodology and data

a) Please provide units for relevant parameters such as α_s etc. to make the values interpretable. If this is in 1/h, SOX2 would have a very long half-life of nearly 20 h, but Liao et al. 2018 showed that the half-life of SOX2 is on the order of 1 h. How would changing α_s in the model to the measured value affect the conclusions?

b) Please explain in more detail why the data should suffer from "noisy high-dimensional single cell measurements". The Introduction appreciates "meaningful heterogeneity" – which could be useful for the interpretation of the data – and it is unclear why the high-resolution images and long time scales in this work should suffer from noise.

c) Does the data predict additional spatial patterns relevant for the biological system if more clusters are chosen in Fig. 1I?

d) The artificial neural network and support vector approaches are not adequately explained, and the code was not provided for evaluation.

e) GSE229675 is currently private and could not be evaluated (but the analyzed data is available on GitHub).

f) Please submit the code to Zenodo to preserve the version and ensure transparency and reproducibility.

g) Please also provide the raw images in a data repository such as Zenodo.

4) General

a) The authors frequently mention predictions, but much of the current work is based on correlations rather than direct functional tests (e.g. line 168 etc.). Such statements should be toned down.

- b) Please also discuss and cite relevant previous work with similar observations, such as Dubrulle et al. 2015.
- c) The Introduction should provide more background and better explain what exactly is controversial in the cited studies to guide the readers.

MINOR POINTS

- a) Some content in the manuscript has been mixed up, which makes it hard to read (e.g. the legend for Fig. 3 lists violin plots, but these are not shown etc.). Please carefully double-check all references to figures to enhance readability.
- b) It looks as if some of the data is cut off below the x axis in Fig. S4G. Please clarify.
- c) Please re-position the legend in Fig. S5H and Fig. 5I, so that the curves at later time points are visible.
- d) The lists mentioned on p. 14 should be provided as Supplementary Information.
- e) Please include additional labels in Fig. 3L, so that the conditions can be distinguished.
- f) Nie et al. 2014 in the Methods section does not seem to appear in the References.
- g) It would be ideal to repeat the inducible SOX2 experiment in micropattern cultures.
- h) It would be ideal to assess the stability of BMP4 in cell culture by probing the cell culture supernatant over the course of 40 h. This might give an explanation for the sigmoid signaling profiles.

Reviewer #2 (Remarks to the Author):

The paper by Teague et al investigates how BMP signaling in human ES cells (hESCs) determines cell fate. The authors mainly investigate the fate choice between pluripotency and amnion fate. The authors develop a nice assay whereby they can track BMP signaling over time in individual cells using nuclear localization of SMAD4 as a readout and then assay cell fate with specific markers by immunostaining at the end of the assay. They show that the important determinant of cell fate choice is the integral of BMP signaling rather than signaling duration or levels of signaling. They look for a gene that might act as an integrator of signaling and be responsible for these observations. They find SOX2 as a candidate and show that SOX2 overexpression can inhibit differentiation.

In general, I think that the work is of a high standard and makes some very interesting observations, which will be important for the field. I have some comments and questions as outlined below.

1. At the beginning of the paper the authors investigate both BMP and Nodal signaling. The obvious clean readout of Nodal signaling is SMAD2 nuclear localization. The authors should test this in addition to SMAD4 and SMAD1 (for BMP signaling).
2. Since SMAD1 is directly controlled by BMP signaling I would have assumed that the results with SMAD1 nuclear localization would be at least as good as SMAD4. In line 246, the authors state that the results with SMAD1 are much noisier. Why is this?

3. The effects of adding exogenous BMP will be confounded by the production of BMPs by the cells. To what extent do these cells produce BMPs? This needs to be investigated.

4. The authors need to cite PMID 32202522 in line 371.

5. For the experiment shown in Figure 5A, the stability of the proteins is going to confound the kinetics of protein production with different levels of BMP signaling. The authors need to investigate this.

6. I am not very convinced about the identification of SOX2 as the integrator. Their evidence is correlative. Overexpressing SOX2 and showing this inhibits BMP-induced differentiation is not conclusive evidence. What happens if other pluripotency transcription factors like NANOG and OCT4 are overexpressed? Much stronger evidence is required for the authors to be able to conclude that SOX2 is the integrator.

7. In the discussion the authors state that the integrator must be infinitely stable (line 522). This cannot be the case for a protein like SOX2 whose expression is repressed by BMP signaling. In order to see repression, it must be very unstable.

Small points.

In line 138, I don't think that the authors mean to cite SI Fig 1I here.

In Figure 5B the authors state "level of SMAD4", but they mean level of nuclear SMAD4. Same point for line 452

Reviewer #3 (Remarks to the Author):

In this manuscript, Teague and colleagues address a fundamental question: how does morphogen signaling regulate cell fate. The authors have developed an automated tracking method to follow signaling histories linked to specific cell fates in hPSCs. Using an unbiased statistical approach, they show that the BMP signaling history in individual cells, and in particular the time integral of signaling, correlates strongly with cell fate - pluripotent vs amnion like. This is an important finding that provides an alternative to models in which the duration or levels alone are interpreted to define cell fates. The finding opens new questions about the mechanisms by which cells measure signaling integrals. The authors suggest that in this system this is achieved by tuning the levels of Sox2. This is overall an interesting and clear manuscript. I have a few, mostly minor comments that should be addressed prior to publication.

Major comment:

1. The temporal dynamics presented in Figures 1F and 1K appears very different to what is shown in Fig. 3H. It is not clear where this difference comes from.

The profiles of the cells Figure 1L should be shown in a similar manner to those of 1H to demonstrate that these data are similar to 3H, or a clarification should be provided.

Minor comments:

There are multiple inaccuracies in calling the figures. All figure references should be rechecked.

Figure 1F, heat map key is reversed such that red is low and high is blue, this is confusing and the opposite to all other figures including the adjacent 1E. This is also the case is SI.Fig 4Q.

Line 138: Reference to SI Fig 1F or G instead of I?

Line 142 “sustained BMP signaling leads to amnion-like differentiation”. Later in the paper however it appears that, in the Wnt inhibited data, transient high BMP seems sufficient to induce amnion-like cell fates – the authors should clarify this.

Line 158: what is the difference between Fig 1L and SI Fig 1J aside from the Y axis? Please provide more clarity in either the text or legend as the legend entries are identical.

Line 161: Reference to SI Fig 1L in text?

There is no reference in text to SI Fig 3G.

Fig 3I,J,M,N,O log (ISL1/Nanog) are inconsistent between heatmaps and scatter plots. Maybe 1 should always be higher than -1 for consistency?

In Fig 3L, the conditions need to be indicated in the plot so that one can visualize how mean signaling history and fate varies by condition. Fig 3L+N, SI Fig 3P, the conditions need to be properly described in fig legend. The figure legend currently refers to Q, which does not exist.

Fig 3 has two P, one of which is in the wrong place (line 310) and describes M. M (line 311) meanwhile references violin plots which are not present in this figure. All figure legends need checking to ensure accuracy.

In Fig 3P, please provide full forms for NN and SVM at least in the legend for P. Also include in methods about these classifiers?

Line 357: Is this again with Wnt inhibition, please clarify.

Line 371: include reference.

Line 424: SI Fig 4F, right?

Line 439: Fig 3I? Did you mean 3G?

Line 450 references to SI Fig. 5G should be Fig. 5H and I?

Line 457: please give ref to fig.

Line 461: also refer to Fig 5K?

Line 1117: 'as in Fig 5K', instead of 4K?

In Fig 5, you always relate Sox2 expression with SMAD4 levels, but what about SMAD4 integral? Does the decay in Sox2 expression also correlate with the integral of BMP signaling?

Line 525: reference in Fig?

Line 559: SI Fig 4R, right?

In Fig 5L, M, over expression of Sox2 is found to reduce differentiation to amnion-like fate in response to BMP. Does the over expression of SOX2 lead to a change in the integral threshold for amnion cell fate specification?

RESPONSE TO REVIEWERS

Please note references are at the end of our response.
Reviewer comments are *italicized* with our response below.

Reviewer #1:

In this manuscript, Teague et al. investigate the relationship between single cell signaling histories and cell fate, focusing on the decision between amnion-like and pluripotent cells controlled by BMP in the absence of WNT. The authors developed an imaging pipeline with automated tracking and impressive multiplexed immunofluorescence stainings, resulting in a large dataset of signaling histories linked to cell fate. They conducted experiments with different signaling levels and durations, and observed that the duration threshold for differentiation changed depending on the signaling level, but the integrated signaling remained the same. The authors then explored possible mechanisms by which cells might integrate BMP signaling and identified SOX2 as a potential integrator. SOX2 levels were found to reflect integrated SMAD4 signaling in an inverse relationship, and a simple model incorporating SOX2 production, degradation, and negative regulation by ISL1 showed good agreement with the experimental data. Finally, using a doxycycline-inducible SOX2 cell line, the authors demonstrated that maintaining SOX2 levels by brief doxycycline treatment prevented upregulation of amnion markers.

Overall, the work is relevant for the fields of development and stem cell biology, but the finding that both signaling level and duration are important for morphogen interpretation is not new and has been demonstrated in multiple systems. Several predictions in the current manuscript are based on correlations rather than direct tests, and additional evidence is needed to substantiate the claims and resolve conflicts with other studies as detailed below.

We thank reviewer 1 for their thorough review and detailed comments. We believe we were able to address most if not all concerns. Regarding the big picture, we would like to point out that while the importance of duration and level was appreciated before, its implications for BMP signaling in human pluripotent stem cell differentiation were unknown. More importantly, a quantitative relationship between BMP signaling level, duration and differentiation had remained elusive in any system and discovering it was only possible in this context due to our significant experimental innovations. We did not simply find duration and level are important, we demonstrated that both are interpreted quantitatively through the time integral, which is a new and non-trivial finding that goes significantly beyond establishing importance of these variables.

Our finding that cell fate heterogeneity in stem cell differentiation strongly correlates with heterogeneity in signaling history in figure 3 is itself an important and highly non-trivial finding, as the source of heterogeneous differentiation in response to uniform exogenous ligands remains a subject of much debate. However, going beyond correlation, we manipulated level and duration independently in figure 4 to get data that we think clearly rule out the alternative hypothesis. In addition, our work emphasizes an important difference between the supplied concentration of an exogenous ligand and the actual level of cell signaling in response (i.e. the concentration of BMP affects duration of signaling response more than level).

We acknowledge the fact that our data on SOX2 as an integrator is not conclusive and the mechanism of integration is an ongoing area of research in the lab. At the request of the reviewers, we now provide additional evidence that SOX2 may be an integrator gene, including a revised model consistent with the SOX2 half-life we measured, and additional evidence for direct regulation of SOX2 by BMP and ISL1 by SOX2. However, we only meant to demonstrate that SOX2 is a strong candidate, while acknowledging that the full gene regulatory network involved is likely much more complex and will require several more manuscripts worth of work to be revealed. We have changed our wording to reflect this. We believe we conclusively proved that cells integrate BMP signaling, and we show that all our data can be explained by a simple mechanistic model for integration. With SOX2 we demonstrate that there are genes that behave as expected by this model. Finding such genes would not be expected in a scenario where signaling level thresholds control gene expression, thus strengthening our conclusions.

MAJOR POINTS

1) Readout of BMP signaling

a) There are concerns regarding the choice of SMAD4 as a readout for BMP signaling. The SMAD1 and SMAD4 graphs in Fig. 1E and Fig. S1D indicate that SMAD4 is not a good readout for BMP activity since SMAD1 is expressed uniformly until about 17 h (Fig. S1D,H), whereas SMAD4 is only expressed uniformly until about 11 h (Fig. 1E).

The time at which initially uniform BMP signaling is restricted to the edge reflects epithelial polarization and inhibitor production which depends on initial cell density¹, and varies from cell line to cell line, for example due to growth rate differences. To demonstrate that the difference between Fig 1E and S1D,H (now S1D,J) is not due to a discrepancy between SMAD1 and SMAD4, but rather due to slightly different colony dynamics in Fig 1E vs. S1D,J, we carried out new experiments in which we mixed the RFP::SMAD1 and GFP::SMAD4 cell lines in the same colony so that they are subject to the same

colony-level effects (new SI Fig. 1T-W). In support of our interpretation, we found that in this case, SMAD1 and SMAD4 dynamics match. We now discuss this further in the main text.

More generally, we point out that the validity of SMAD4 as a readout for BMP signaling in hESCs has been established in detail in previously published work, including Nemashkalo et al 2017², Yoni et al 2018³ and Heemskerk et al 2019⁴. For example, Heemskerk et al 2019 Fig 1h and SI Fig. 1f respectively show that GFP::SMAD4 response to BMP4 is not significantly affected by TGF- β receptor inhibitor SB431542 and that the dynamics of pSmad1/5/9 after BMP treatment match those of GFP::SMAD4.

We performed several additional experiments here to further strengthen this point:

- New SI Fig. 2LM show that baseline signaling before BMP treatment in GFP::SMAD4 cells is not affected by SB431532.
- New SI Fig. 2C,H shows the same dose-dependent dynamics in GFP::SMAD4 and RFP::SMAD1 in disordered cells treated with BMP4 and Wnt inhibitor
- New SI Fig 2NO show pSMAD1/5/9 staining matches nuclear GFP::SMAD4 spatially and temporally after treatment with different doses of BMP4.
- New SI Fig. 4AB show pSMAD1/5/9 staining matches nuclear GFP::SMAD4 when BMP signaling level is manipulated with the receptor LDN193189 under sparse culture conditions.
- New SI Fig. 4C shows RFP::SMAD1 dynamics under these conditions match GFP::SMAD4 in Fig. 4G.
- New SI Fig. 4EF show that the integral but not the level of pSMAD1/5/9 (constructed from a time series of stainings) predicts differentiation, matching the conclusions from GFP::SMAD4.
- New SI Fig. 4G-J repeat the SMAD4 integral versus level and duration experiments in the presence of TGF-beta receptor inhibitor SB431542 (SB) and again reach the same conclusion.

These experiments are discussed further below.

b) The role of Nodal in these experiments should be tested conclusively with a Nodal inhibitor, particularly given the findings by Guglielmi et al. 2021 regarding the role of SMAD4 to mediate Nodal and BMP signaling.

We are not sure if “these experiments” refers to the experiments described in a) or all experiments in the manuscript. If a), we hope the response above addresses this.

We did not perform experiments with Nodal (TGF- β) signaling inhibitor in the original manuscript because TGF- β inhibition destabilizes the pluripotent state. That means we would not only be looking at the effect of BMP on differentiation, but also the effect of loss of Nodal signaling, making it more difficult to understand the role of BMP. Rather than staying pluripotent or becoming amnion-like, the presence of TGF- β inhibitor cells would put cells on a path to ectoderm differentiation which could affect amnion competence in response to BMP in an uncontrolled manner. Therefore, we performed our experiments in conditions where constant low levels of exogenous TGF- β maintain pluripotency and only manipulated BMP signaling.

However, we have now repeated the key experiment from Fig. 4 in the presence of Nodal inhibitor SB and found the results in support of our conclusion that BMP signaling is integrated over time (new SI Fig. 4G-J). We believe this significantly strengthens our conclusions since NANOG is lost in the presence of SB, meaning that the prediction of differentiation to ISL1+ amnion-like cells is still accurately predicted by integrated BMP signaling even when the remaining cells lose pluripotency and differentiate to ectoderm.

Moreover, we now show that the background levels of TGF- β signaling are not detectable with GFP::SMAD4 since adding SB does not lower the GFP::SMAD4 nuclear levels in base medium (new SI Fig. 2LM). Thus, any measurable increase in nuclear SMAD4 must be due to BMP. The fact that for example Fig 4 shows a direct relationship between the level of BMP receptor inhibitor and the level of SMAD4 signaling also supports that. Nevertheless, background TGF- β may slightly inhibit BMP signaling, as GFP::SMAD4 response to BMP is slightly increased in the presence of SB (new SI Fig. 4GH), consistent with work by other groups⁵. Notably, this slight increase in BMP signaling in the presence of SB does not affect the quantitative relationship between BMP signaling and differentiation, as seen in new SI Fig. 4J.

We now also explicitly show in micropatterned colonies that Nodal expression is severely reduced upon treatment with Wnt inhibitor, and that this leads to a large reduction in SMAD2/3 signaling without affecting SMAD1 signaling (new SI Fig. 1I-M), further supporting that endogenous Nodal does not play a significant role in the presence of Wnt inhibitor. In addition, we again refer to the controls performed in Heemskerk 2019. Furthermore, the role of Nodal signaling and inhibition of Nodal signal at different times in micropatterned colonies was dissected in great detail in Chabbra et al⁶.

In summary, the conditions we chose allow us to isolate the relationship between BMP signaling and differentiation by allowing low levels of exogenous TGF- β but eliminating endogenous Nodal. Nevertheless, we have now shown that the relationship between integrated BMP signaling and differentiation still holds when TGF- β signaling is inhibited.

c) *Given these concerns, it will be important to repeat the key findings of the current work with the SMAD1-RFP cell line to bolster the conclusions.*

As described above, we have now verified in several places that GFP::SMAD4 signaling dynamics match pSMAD1/5/9 staining and RFP::SMAD1 signaling when Wnt is inhibited. Unfortunately, we found that repeating key differentiation experiments with RFP::SMAD1 cells was not feasible as these cells show severely reduced differentiation compared to GFP::SMAD4 and wild type cells (new SI Fig. 2FG). This suggests that function of RFP::SMAD1 is compromised. Nevertheless, SI Fig. 3L shows that like for GFP::SMAD4, the mean duration of RFP::SMAD1 signaling in differentiated cells is longer than in undifferentiated cells. We resorted to time series of pSMAD1/5/9 staining to further bolster our conclusion that integrated BMP signaling causes differentiation:

- New SI Fig 2NO show pSMAD1/5/9 staining matches nuclear GFP::SMAD4 spatially and temporally after treatment with different doses of BMP4.
- New SI Fig. 4AB show pSMAD1/5/9 staining matches nuclear GFP::SMAD4 when BMP signaling level is manipulated with the receptor LDN193189 under sparse culture conditions.
- New SI Fig. 4EF show that the integral but not the level of pSMAD1/5/9 (constructed from a time series of stainings) predicts differentiation, matching the conclusions from GFP::SMAD4.

d) *The claim “In other words, different concentrations of BMP lead to different signaling durations but not different signaling levels” needs to be corroborated by checking phosphorylated SMAD1/5/9.*

We show that as for GFP::SMAD4, initial response of pSMAD1/5/9 is high and nearly dose independent, while response decreases in a dose-dependent manner later (new SI Fig. 1NO). We further see similar BMP4 dose-dependent signaling dynamics in GFP::SMAD4 in SI Fig. 2C and in RFP::SMAD1 in new SI Fig 2H. As described above, we now also show strong correlation between nuclear:cytoplasmic GFP::SMAD4 and phosphorylated SMAD1/5/9 in new SI Fig. 4AB. Overall this supports our conclusion that there is no concentration of BMP in the media that can create a steady intermediate signaling response. This is important to know for directed stem cell differentiation.

We also reworded the statement “consistent with (SI Fig. 2C), varying BMP concentration and cell density changed the duration of the response but led to minimal variation in the initial and final signaling levels (Fig. 3L).”

e) *If projected on a different color range, the tails in Fig. 1E and Fig. S1D,H at $t > 40$ h might look similar, arguing for a more complicated signaling landscape. Please clarify by ruling out potential normalization issues, and explore different color ranges/thresholds to address this concern.*

We have now plotted the profiles with and without WNT inhibitor at 42h with normalization from 0 to 1 to allow direct comparison of GFP::SMAD4 and RFP::SMAD1 in new SI Fig. 1KL. In addition, we have added a SMAD2/3 staining (new SI Fig. 1MN). This shows that the tail in GFP::SMAD4 looks different from RFP::SMAD1 and is severely reduced when Nodal is lost upon WNT inhibition, and that SMAD2/3 undergoes a similar reduction, but that RFP::SMAD1 is identical between these conditions.

In addition we refer to the stainings of pSmad1/5/9 and SMAD2/3 over time in Heemskerk et al 2019⁴ which show BMP signaling restricted to the edge while SMAD2/3 extends further inward in a manner that matches GFP::SMAD4, consistent with the difference between RFP::SMAD1 and GFP::SMAD4 we show here.

2) Integration model

a) *The paper claims that BMP signaling is integrated over very long times scales of more than 40 hours. This strong claim needs more conclusive tests. If the signal is truly integrated, recurring hour-long pulses of inhibition and strong activation should give the same outcome in terms of amnion differentiation as the an integral of the same magnitude that provides continuous activation at a lower level. This additional test of the integration model is also important to support the authors’ statement that “Integration reduces noise and is insensitive to brief signaling perturbations in the same way as averaging”.*

We performed experiments with pulses in new Fig. 4IJ demonstrating that the differentiation after multiple pulses of BMP signaling matches that of a single pulse with the same time integral.

We want to point out that pulsing experiments are difficult to do since hPSCs are very sensitive and we do all our manipulations manually. 1h pulses were not feasible experimentally since on the one hand it would both perturb the cells too much from opening the incubator and washing out the inhibitor with high frequency and on the other hand it would require lab members to stay up through the entire night. Moreover, it is very hard to wash out inhibitor completely or to shutdown BMP signaling by BMP removal alone (i.e. without inhibitor), thus making many repeated pulses of equal height impossible for us to achieve.

Further, we want to emphasize that we expect integration is an approximation because any integrator must have finite turnover, as we wrote the section “BMP signaling may be integrated by SOX2”, and our mathematical model includes finite turnover (half-life of 7h, new SI Fig. 5IJK) for the integrator protein. However, integration is a very good approximation since under the conditions we tested, the finite-turnover model approximates the true integral within the error bars. Moreover, any deviations from the integral due to turnover would make it a weighted integral (i.e. something slightly more complicated), which nuances our conclusion but does not essentially change it, and still is fundamentally different from fixed level and duration thresholds, which our data are not consistent with.

b) The paper overlaps to a good degree with the preprint from Camacho-Aguilar et al. 2023 who also show integration in time and state that “SOX2 decay dynamics show a very high correlation with the integral of SMAD4 in time“. However, the conclusion of the paper “We showed a BMP-induced morphogen effect in the duration but not the concentration of signaling, indicating that duration and concentration of BMP signal are not interchangeable in this context“ is different from the conclusions of the current work. It will therefore be necessary to more conclusively test the integration model with the pulses described above to resolve this conflict.

We are aware of concurrent work by Camacho-Aguilar. Indeed, they also observed a correlation between SOX2 levels and the SMAD4 integral (in fact this way of phrasing it is based on discussion with us). In the discussion of our manuscript, we explain how their work is largely distinct from ours and consistent with it. Namely, we look at BMP signaling in the presence of Wnt inhibitor and track this in individual cells, whereas they look in bulk at the expression of endogenous Wnt downstream of BMP and how Wnt signaling subsequently interplays with BMP – it is because of this interplay with Wnt that duration and level of BMP are not equivalent for them. The context is thus different. However, consistent with our findings they also find that concentration of BMP translates primarily to duration of signal response. We have discussed this with them, they agree there is no contradiction, and they will also include this in their discussion.

c) The prediction that overexpression of SOX2 reduces differentiation to amnion-like fate in response to BMP is not a strong one and appears to be compatible with multiple morphogen interpretation models, not exclusively the integration model. If the authors disagree, they should more explicitly describe in the text why the outcome of the experiment is incompatible with other morphogen interpretation models.

Our data in figure 4 rules out morphogen interpretation models based on fixed level or duration thresholds, independent of downstream mechanism. We then look for a potential mechanism for integral interpretation. The rate of SOX2 decay is linear in the level of BMP signaling. This continuous linear dependency is non-trivial behavior and, in some sense, the opposite of the sharp transition from on to off at a specific level that is expected for models involving level thresholds. In other words, it is the quantitative relationship between SOX2 and BMP signaling that supports the integral model, not the overexpression phenotype.

The fact that its overexpression reduces differentiation then suggest SOX2 has a functional role rather than being only a correlate of the integral. We now repeated this experiment in micropatterned colonies and included GATA3 staining (new Fig. 5NO, new SI Fig. 5Q). Consistent with our model, SOX2 overexpression reduced ISL1 but not GATA3. We expect the full gene regulatory network downstream of BMP to be more complex and there may be multiple integrators working in parallel, but we believe showing the existence of at least one integrator gene, as defined by levels matching the integral of BMP signaling and direct response to BMP (in bulk RNA-seq, but also see ChIP-seq below), does strengthen support for our model. We have clarified the text to make this clearer and more nuanced.

d) The authors selected SOX2 as an integrator candidate, but the evidence is not strong. Additional data would strengthen this hypothesis. For example, what is the experimental evidence that SOX2 directly represses ISL1, and does ISL1 repress SOX2? Is SOX2 a direct target of pSMAD, and does pSMAD bind in SOX2 promoter/enhancer regions?

Several publications that we now cite in the manuscript support direct regulation of SOX2 by pSMAD1⁷⁻⁹. In addition, we have analyzed existing SOX2 and pSMAD1 ChIP-seq data and found overlapping binding peaks of SOX2 and pSMAD1 on active ISL1 enhancers, as well as on the SOX2 promoter (new SI Fig. 5L). However, even if SOX2 repressed ISL1 indirectly through an intermediate, it could still perform the same function of integrating BMP to control differentiation.

3) Methodology and data

a) Please provide units for relevant parameters such as alpha_s etc. to make the values interpretable. If this is in 1/h, SOX2 would have a very long half-life of nearly 20 h, but Liao et al. 2018 showed that the half-life of SOX2 is on the order of 1 h. How would changing alpha_s in the model to the measured value affect the conclusions?

We thank the reviewers for this important question. We have added units and indeed, SOX2 needs to have a long half-life for it to perform the role of integrator in our original model. However, we have found that a modified model including SOX2 autoregulation explains our data with a shorter SOX2 half-life.

SOX2 half-life in mouse ESC is most likely around 8h. In contrast to the 1h half-life found by Liao 2018, Liu 2017 found a half-life of around 8h¹⁰. Both papers rely on quantifying western blot after cycloheximide treatment, which is not the most reliable method as cycloheximide perturbs all the machinery of the cell by blocking translation and in our experience rapidly kills cells. Measurement of protein turnover in steady state in live cells by photobleaching, photoconversion, or transient fluorescent labeling using e.g. SNAP-tag is more reliable (although like every method these have their own shortcoming, since the tag might change the protein half-life). SNAP-tag measurements of SOX2 half-life in mESC yielded a half-life around 8h^{11,12}, supporting the number found by Liu et al.

To determine SOX2 half-life in human PSCs, we measured GFP::SOX2 fluorescence recovery after photobleaching, yielding a half-life around 7h, similar to mouse ESCs (new SI Fig. 5IJK). This was inconsistent with our original model. However, it is well understood in systems biology that autoregulation can decouple the timescale for the system to reach equilibrium from the protein half-life and that positive autoregulation slows the dynamics down¹³. Moreover, SOX2 is known to regulate itself¹⁴, and we now include ChIP-seq analysis further supporting this (new SI Fig. 5L). Therefore, we incorporated positive SOX2 autoregulation in our model and found that it fit our data with the measured half-life.

b) Please explain in more detail why the data should suffer from “noisy high-dimensional single cell measurements”. The Introduction appreciates “meaningful heterogeneity” – which could be useful for the interpretation of the data – and it is unclear why the high-resolution images and long time scales in this work should suffer from noise.

Signaling histories are high-dimensional data because they contain measurement for many different time points. Single cell signaling measurements using live cell imaging are generally noisy as can be seen not only in the single cell signaling traces in our work (e.g. Fig. 3A, SI Fig. 3CD), but also other work on single cell signaling dynamics, e.g.^{15,16} To be more explicit about this, we have quantified variance in our live-cell measurements in new SI Fig. 2I-K. As one source of noise for us, expression levels of fluorescent fusion proteins under endogenous control are relatively low, leading to lower signal-to-noise images with light exposures low enough to prevent phototoxicity. Other sources of variability in the signal for individual cells include changes in the z-position of the nucleus which affects how cleanly nuclear fluorescence is separated from cytoplasmic fluorescence, as well as other changes in cell shape that affect the amount of SMAD4 in one point spread function volume, and local differences in background fluorescence, junk, and dead cells nearby.

We appreciate the question of how one makes the distinction between noise and meaningful heterogeneity. We define meaningful in this context as containing information about differentiation. We found that denoising improves correlation between signaling and cell fate markers (Fig. 3JN, SI Fig. 3F,M-P), which is what is expected from removing noise. The opposite would be expected if heterogeneity informative of cell fate were removed by this operation.

c) Does the data predict additional spatial patterns relevant for the biological system if more clusters are chosen in Fig. 1I?

We chose 3 clusters because the principal component analysis in Fig. 1G clearly shows three parts to the signaling data. We have now included the patterns for different cluster numbers in new SI Fig. 1H, but it is not clear to us if these are biologically relevant.

d) The artificial neural network and support vector approaches are not adequately explained, and the code was not provided for evaluation.

We apologize that this was unclear, we have added more explanation in the methods section. The code was provided at the public github repository github.com/idse/BMPintegral under `figures/single_cell_history_analysis/` (the home page describes where to find the code for each figure). However, to make the code clearer we have cleaned it up and added additional comments.

e) GSE229675 is currently private and could not be evaluated (but the analyzed data is available on GitHub).

We usually make data public upon publication, but we have now made it public so the reviewer can already access it.

f) Please submit the code to Zenodo to preserve the version and ensure transparency and reproducibility.

For all our past work we have preserved the final version of our code related to that paper on Github as a separate repository which we no longer alter (e.g., <https://github.com/idse/PGCs> and <https://github.com/idse/FatDs>) and will do the

same here. At your suggestion we have also preserved the preprint code on Zenodo at <https://doi.org/10.5281/zenodo.10076773>.

If any more changes are made will also preserve the final version of the code on Zenodo upon publication.

g) Please also provide the raw images in a data repository such as Zenodo.

This poses a practical challenge. High resolution live imaging generates about 200GB of raw data per experiment (including segmentation and stitched tilings will be even larger). As such the total raw data size for the paper figures is on the order of 3TB. For Zenodo "Total files size limit per record is 50GB". (<https://about.zenodo.org/policies/>) However, we are always happy to share live data upon request and we hope this is sufficient.

4) General

a) The authors frequently mention predictions, but much of the current work is based on correlations rather than direct functional tests (e.g. line 168 etc.). Such statements should be toned down.

This may be a semantic issue, but we use prediction in the statistical sense: if I give you the value of one variable, can you give me (predict) the value of the other? Correlation therefore enables prediction, but a causal relationship is not implied. In our experience this is how the word is most frequently used, also in the quantitative embryonic patterning literature^{1,17,18}. To avoid confusion, we have changed our word choice in many places and added a footnote to clarify what we mean by prediction.

b) Please also discuss and cite relevant previous work with similar observations, such as Dubrulle et al. 2015.

Thank you for the suggestion, we have included Dubrulle 2015 in our discussion.

c) The Introduction should provide more background and better explain what exactly is controversial in the cited studies to guide the readers.

We thank the reviewer for this comment. While detailed discussion would take up too much space, we have edited the text to make clear that the papers do not agree about whether a gradient-threshold model explains expression domains of BMP target genes.

MINOR POINTS

a) Some content in the manuscript has been mixed up, which makes it hard to read (e.g. the legend for Fig. 3 lists violin plots, but these are not shown etc.). Please carefully double-check all references to figures to enhance readability.

Thank you for pointing this out, we have corrected it.

b) It looks as if some of the data is cut off below the x axis in Fig. S4G. Please clarify.

Thank you for pointing this out we have corrected it.

c) Please re-position the legend in Fig. S5H and Fig. 5I, so that the curves at later time points are visible.

We have done this.

d) The lists mentioned on p. 14 should be provided as Supplementary Information.

Thanks for pointing this out, we have now included a Supplementary Table 1 as a file called integratorgenecandidates.csv which contains the cluster assignment, slope of transcription, and correlation with SMAD4, as well as their status as a potential integrator based on the values for each gene that undergoes significant fold change during differentiation, and whether they are transcription factors (based on¹⁹). We point out that there are minor changes in the number of genes in each group after intersecting the genes present in both the time series and dose response at the start of the analysis.

e) Please include additional labels in Fig. 3L, so that the conditions can be distinguished.

It became cluttered in the main figure with all the extra labels, so we have included a supplementary figure with these labels (SI Fig. 3N).

f) Nie et al. 2014 in the Methods section does not seem to appear in the References.

Thank you for pointing this out, we have corrected it.

g) It would be ideal to repeat the inducible SOX2 experiment in micropattern cultures.

We have done this experiment; the results are in new Fig. 5NO and new SI Fig. 5Q and were consistent with the experiment in standard culture. This time we also stained for GATA3 and consistent with our model SOX2 overexpression reduces ISL1 but not GATA3.

h) It would be ideal to assess the stability of BMP4 in cell culture by probing the cell culture supernatant over the course of 40 h. This might give an explanation for the sigmoid signaling profiles.

Thank you for this suggestion. We performed this experiment and found that the supernatant of cells treated with 50ng/ml BMP4 in which signaling has gone down still elicits a strong response in previously untreated cells, while treatment of the original cells with fresh BMP4 no longer elicits response, so that BMP4 degradation does not explain the sigmoid profile under these conditions (new SI Fig. 2DE).

However, we previously fitted a model where BMP is degraded after receptor binding to BMP dose response data in Heemskerk Elife 2019. Importantly, under those conditions, density was very low and response to high doses of BMP was maintained as expected from the model. In SI Fig. 2 and Fig. 3 in this work we chose higher cell densities to get more heterogeneity in signaling and cell fate. Under these conditions, signaling is reduced at later times and sigmoidal in shape even at high doses. We noticed that this coincides with colonies merging and therefore colony borders disappearing. Inhibition of BMP signaling is to be expected in this case based on Etoc et al 2016, which demonstrated strong BMP response is restricted to colony edges by both receptor polarization and inhibitor secretion. Thus, there are likely multiple independent effects that reduce BMP signaling over time.

We feel that it is important to point out that our conclusion that cells integrate BMP signaling does not depend on the sigmoid behavior. In fact, experiments in Fig. 4 were under low density conditions where there is no sigmoid behavior. More generally, to test the hypothesis that signaling history explains differentiation, the upstream mechanisms that generate the signaling dynamics should not matter. We have revised the text to clarify this.

Reviewer #2:

The paper by Teague et al investigates how BMP signaling in human ES cells (hESCs) determines cell fate. The authors mainly investigate the fate choice between pluripotency and amnion fate. The authors develop a nice assay whereby they can track BMP signaling over time in individual cells using nuclear localization of SMAD4 as a readout and then assay cell fate with specific markers by immunostaining at the end of the assay. They show that the important determinant of cell fate choice is the integral of BMP signaling rather than signaling duration or levels of signaling. They look for a gene that might act as an integrator of signaling and be responsible for these observations. They find SOX2 as a candidate and show that SOX2 overexpression can inhibit differentiation.

In general, I think that the work is of a high standard and makes some very interesting observations, which will be important for the field. I have some comments and questions as outlined below.

We thank the reviewer for these kind words.

1. At the beginning of the paper the authors investigate both BMP and Nodal signaling. The obvious clean readout of Nodal signaling is SMAD2 nuclear localization. The authors should test this in addition to SMAD4 and SMAD1 (for BMP signaling).

We and others previously looked at SMAD2/3 nuclear localization^{4,6}. That work explained the SMAD4 dynamics as a superposition of the SMAD2/3 and SMAD1/5/9 dynamics. To further support this and show the change in SMAD2/3 signaling (absence of a Nodal wave) in the presence of the Wnt inhibitor iwip2, we have performed SMAD2/3 immunofluorescence staining with and without Wnt inhibition (new SI Fig. 1MN) which can be compared with RFP::SMAD1 and GFP::SMAD4 in new SI Fig. 1KL). We now also show explicitly that Nodal expression is severely reduced upon Wnt inhibition (new SI Fig. 1I). We only consider Nodal signaling in the first half of Figure 1 and focus on BMP signaling in the remainder of the paper.

2. Since SMAD1 is directly controlled by BMP signaling I would have assumed that the results with SMAD1 nuclear localization would be at least as good as SMAD4. In line 246, the authors state that the results with SMAD1 are much noisier. Why is this?

This is because expression levels of SMAD1 are much lower than SMAD4 so even with laser settings that are bordering phototoxicity we unfortunately get low signal. We have now quantified this in new SI Fig. 2I-K. Careful inspection of Fig. 1C also shows that RFP::SMAD1 levels are much closer to background than GFP::SMAD4, something we were unable to improve with different exposures without affecting cell health. Importantly, in performing the revision experiments we discovered RFP::SMAD1 is likely functionally compromised as these cells do not differentiate well in response to BMP. This provides another reason to focus on SMAD4. We have performed a range of additional controls to verify that the GFP::SMAD4 nuclear:cytoplasmic ratio faithfully reports BMP signaling and matches both RFP::SMAD1 and pSMAD1 immunofluorescence staining:

- New SI Fig. 1T-W shows that GFP::SMAD4 and RFP::SMAD1 dynamics match when these cells are mixed in the same micropatterned colony.
- New SI Fig. 2C,H shows the same dose-dependent dynamics in GFP::SMAD4 and RFP::SMAD1 in disordered cells treated with BMP4 and Wnt inhibitor
- New SI Fig. 2NO show pSMAD1/5/9 staining matches nuclear GFP::SMAD4 spatially and temporally after treatment with different doses of BMP4.
- New SI Fig. 4AB show pSMAD1/5/9 staining matches nuclear GFP::SMAD4 when BMP signaling level is manipulated with the receptor LDN193189 under sparse culture conditions.
- New SI Fig. 4C shows RFP::SMAD1 dynamics under these conditions match GFP::SMAD4.
- New SI Fig. 4EF show that the integral but not the level of pSMAD1/5/9 (constructed from a time series of stainings) predicts differentiation, matching the conclusions from GFP::SMAD4.

3. The effects of adding exogenous BMP will be confounded by the production of BMPs by the cells. To what extent do these cells produce BMPs? This needs to be investigated.

Amnion-like cells do express BMP4, as can be found in the single cell RNA-sequencing data we published in Jo et al Elife 2022. Importantly, however, our approach measures BMP signaling activity, downstream of both exogenous and endogenous BMP, and our manuscript focuses on the relationship between signaling activity and differentiation. This is in fact a strength of our approach: a variety of effects including endogenous BMP may cause heterogeneous signaling activity, and by relating heterogeneous signaling activity to cell fate we can leverage this heterogeneity to infer the relationship between signaling and fate, regardless of its source. We have edited the section discussing SI Fig 2. and the discussion to make this more explicit.

4. The authors need to cite PMID 32202522 in line 371.

We have now cited Gunne-Braden et al (32302522) on line 371.

5. For the experiment shown in Figure 5A, the stability of the proteins is going to confound the kinetics of protein production with different levels of BMP signaling. The authors need to investigate this.

We thank the reviewer for this important question. This and related questions by the other reviewers made us realize that what matters is whether the total protein level reflects the integral of BMP signaling – this level is the cell memory, and this is what we measured. How this total level depends on protein turnover is then a separate but interesting question. While slow turnover is one way to achieve long memory, as we discuss further below, for GFP::SOX2 we found that protein turnover is faster than the timescale on which SOX2 goes down due to BMP signaling and this can be explained by positive autoregulation. Thus, the memory of the system is longer than the half-life of proteins that constitute this memory.

To directly address the reviewers comment we attempted to measure half-lives of the proteins in 5A by treating with cycloheximide and performing time series immunofluorescence staining but unfortunately found that cycloheximide killed the cells faster than the protein level was decreasing. We also attempted to measure protein production in the presence of the proteasome inhibitor MG132, but this also proved too toxic on the relevant timescales.

6. I am not very convinced about the identification of SOX2 as the integrator. Their evidence is correlative. Overexpressing SOX2 and showing this inhibits BMP-induced differentiation is not conclusive evidence. What happens if other pluripotency transcription factors like NANOG and OCT4 are overexpressed? Much stronger evidence is required for the authors to be able to conclude that SOX2 is the integrator.

We are not claiming SOX2 is *the* integrator, in the sense that it acts alone. However, it is *an* integrator in the sense that it is a direct target of BMP whose levels reflect the time integral of signaling. We have revised our writing to reflect uncertainty regarding this mechanism, e.g. we have changed the section title to “BMP signaling is integrated by SOX2” to “BMP signaling may be integrated by SOX2”. However, we do believe the overexpression shows that SOX2 levels functionally control differentiation rather than only being a correlate of the BMP integral. Strikingly, when we repeated this experiment in micropatterned colonies and included GATA3, we found ISL1 is controlled by SOX2, but GATA3 is not, as we predicted (new

Fig. 5NO). We have now also included ChIP-seq data that suggests direct regulation of ISL1 by a combination of pSMAD1 and SOX2 (new SI Fig. 5L).

We did not overexpress other pluripotency factors and are aware that SOX2 is part of a core pluripotency network with OCT4 and NANOG, so perturbing those will also likely change SOX2 levels. However, we now tested the relationship between integrated BMP signaling and differentiation in the presence of TGF- β receptor inhibitor, which leads to loss of NANOG but preserves the relationship between BMP signaling and ISL1 (SI Fig. 4G-J). This suggests our conclusions do not depend on whether NANOG or pluripotency are maintained. Future work will look in detail at these conditions, which lead to ectoderm differentiation, but it is beyond the scope of the current work.

In summary, we have found a candidate gene consistent with a simple model for integration, which one would not expect to find if signaling were interpreted through a different mechanism like a level threshold. However, we expect the gene regulatory network downstream of BMP to be much more complex and this will be an ongoing subject of investigation in our lab for years to come.

7. In the discussion the authors state that the integrator must be infinitely stable (line 522). This cannot be the case for a protein like SOX2 whose expression is repressed by BMP signaling. In order to see repression, it must be very unstable.

We thank the reviewer for this insightful question. Indeed, intuitively there is some tension between required stability without BMP and the need to decay in the presence of BMP, if – as in our model -- BMP regulates protein production but does not directly regulate protein degradation.

The mathematical model in that sense is essential to show consistency of the hypothesis with the data. The original model shows that although a perfect integrator should be infinitely stable in the absence of BMP, a half-life of 20h is sufficient to explain the observed decay upon BMP treatment and still obtain differentiation downstream of SOX2 that reflects integrated signaling to close approximation. However, we have now measured the half-life of SOX2 and found a half-life of 7h (new SI Fig. 5IJK), which is inconsistent with our original model. We then realized that positive autoregulation by SOX2 slows down the time for the system to reach equilibrium. Including SOX2 autoregulation made the model consistent with our data using the measured SOX2 half-life (new Fig. 5IJK). The model illustrates that the memory of the system can be much longer than the lifetime of its components. Importantly, there is ample evidence for SOX2 autoregulation¹⁴, and we have included ChIP-seq data further supporting this (new SI Fig. 5L)

Small points.

In line 138, I don't think that the authors mean to cite SI Fig 1I here.

Thank you for pointing this out, we have corrected it to SI Fig 1G.

In Figure 5B the authors state "level of SMAD4", but they mean level of nuclear SMAD4. Same point for line 452

Thank you, we corrected 'SMAD4 level' to 'SMAD4 signaling level' throughout the text and to 'SMAD4 (N:C) level' on the graph axis for brevity.

Reviewer #3:

In this manuscript, Teague and colleagues address a fundamental question: how does morphogen signaling regulate cell fate. The authors have developed an automated tracking method to follow signaling histories linked to specific cell fates in hPSCs. Using an unbiased statistical approach, they show that the BMP signaling history in individual cells, and in particular the time integral of signaling, correlates strongly with cell fate - pluripotent vs amnion like. This is an important finding that provides an alternative to models in which the duration or levels alone are interpreted to define cell fates. The finding opens new questions about the mechanisms by which cells measure signaling integrals. The authors suggest that in this system this is achieved by tuning the levels of Sox2. This is overall an interesting and clear manuscript. I have a few, mostly minor comments that should be addressed prior to publication.

We thank the reviewer for the positive comments and careful reading.

Major comment:

1. The temporal dynamics presented in Figures 1F and 1K appears very different to what is shown in Fig. 3H. It is not clear where this difference comes from.

We thank the reviewer for this question. The dynamics in 3H look sigmoidal, going from steady high to steady low. This also happens in the center of the colony in figure 1F,K. What is different is the time of going from high to low and that all cells in

3H end up low while the cells on the edge in 1F,K remain high. Etoc et al 2016 explains what causes this: receptor polarization and secreted inhibitors reduce BMP response in the colony center in a density-dependent manner. The same happens in standard culture but the cell density is lower which makes the response go down slower. Moreover, it eventually reaches confluency, which means there are no edges left and therefore no cells that always maintain high signaling. We now include experiments in new SI Fig. 2DE and a corresponding paragraph to explain this. We emphasize that our manuscript investigates the relationship between signaling and cell fate. What causes heterogeneity in signaling was not subject of investigation and does not impact the conclusions.

The profiles of the cells Figure 1L should be shown in a similar manner to those of 1H to demonstrate that these data are similar to 3H, or a clarification should be provided.

We have added SI Fig. 1Q showing the profiles from 1L in the same manner as 1H. There is some difference between the data in figure 3 and figure 1 that is due to different experimental conditions as explained above. In the micropatterned colonies in Figure 1, the amnion cells on the edge maintain high BMP response (see Etoc et al), whereas in standard culture at densities optimized to give an even mixture of fates, the cells eventually reach confluence, eliminating any exposed edges, so that signaling drops everywhere. This is why even for amnion-like cells the BMP response is not sustained at a high level in figure 3H. This is in fact why the data in figure 3 is more informative – the increased variation in signaling histories provides additional information about how signaling controls fate. We have edited the text further to clarify this.

Minor comments:

There are multiple inaccuracies in calling the figures. All figure references should be rechecked.

We have double checked figure references for the revised manuscript. We apologize if any mistakes still slipped through.

Figure 1F, heat map key is reversed such that red is low and high is blue, this is confusing and the opposite to all other figures including the adjacent 1E. This is also the case is SI.Fig 4Q.

We have corrected this.

Line 138: Reference to SI Fig 1F or G instead of I?

Thank you for pointing this out, we have corrected it to SI Fig 1G.

Line 142 “sustained BMP signaling leads to amnion-like differentiation”. Later in the paper however it appears that, in the Wnt inhibited data, transient high BMP seems sufficient to induce amnion-like cell fates – the authors should clarify this.

Thank you for this question, which is closely related to the major comment addressed above. In fact, this is the main motivation for developing single-cell tracking and exploring different conditions in Fig 2 and SI Fig 2. On the micropattern edge, sustained BMP signaling for 42h leads to amnion-like differentiation while in the interior, 12 hours of BMP signaling does not. It is not clear if there is a duration of BMP signaling in between that is sufficient for differentiation. However, different durations for individual cells arise naturally in disordered culture experiments that we perform in Figures 2 and 3. As we state in text for Figure 2: “The analysis in Fig. 1 suggests that BMP response is high throughout differentiation in future amnion-like cells. However, it cannot be determined whether there is a minimum level or duration of response required for amnion-like differentiation since only a very small range of levels and durations are represented and each history is an average over a cell population.”

Line 158: what is the difference between Fig 1L and SI Fig 1J aside from the Y axis? Please provide more clarity in either the text or legend as the legend entries are identical.

We apologize that this was unclear. Fig. 1L shows cluster assignment based on split 1 in SI Fig. 1S, which is further discussed in the methods section. While SI Fig. 1J (Now SI Fig. 1P) shows assignment based on soft k-means, an automated clustering method. We have revised the captions to emphasize this.

Line 161: Reference to SI Fig 1L in text?

SI Fig. 1L, now SI Fig. 1S illustrates a technicality that we describe in the methods section “Analysis of signaling histories”. We refer to it in the main text and the caption for Fig. 1L now.

There is no reference in text to SI Fig 3G.

We have fixed this. This figure is now SI Fig. 3E.

Fig 3I,J,M,N,O log (ISL1/Nanog) are inconsistent between heatmaps and scatter plots. Maybe 1 should always be higher than -1 for consistency?

Thank you for pointing this out, we have corrected it.

In Fig 3L, the conditions need to be indicated in the plot so that one can visualize how mean signaling history and fate varies by condition. Fig 3L+N, SI Fig 3P, the conditions need to be properly described in fig legend. The figure legend currently refers to Q, which does not exist.

Fig 3L became cluttered in the main figure with all the extra labels, so we have included a supplementary figure with all the conditions labeled (SI Fig. 3N). Thank you for pointing out the lacking legend and mistaken reference to Q, we have fixed this.

Fig 3 has two P, one of which is in the wrong place (line 310) and describes M. M (line 311) meanwhile references violin plots which are not present in this figure. All figure legends need checking to ensure accuracy.

Thank you for pointing this out, we have corrected it.

In Fig 3P, please provide full forms for NN and SVM at least in the legend for P. Also include in methods about these classifiers?

We have revised the caption for Fig. 3P and added a paragraph about the classifiers to the methods section.

Line 357: Is this again with Wnt inhibition, please clarify.

Yes, apologies for not stating this before, we now state this explicitly in the main text and the figure 4P caption.

Line 371: include reference.

We have done this.

Line 424: SI Fig 4F, right?

SI Fig 5F was right, which shows the time course for example genes in the RNA-seq dataset.

Line 439: Fig 3I? Did you mean 3G?

Yes, thank you.

Line 450 references to SI Fig. 5G should be Fig. 5H and I?

We have updated the order in which figures showing GFP::SOX2 data are referenced for clarity.

Line 457: please give ref to fig.

We have added a reference to fig. 5H.

Line 461: also refer to Fig 5K?

Yes, that should have referred to Fig. 5K. However, we significantly revised this paragraph based on comments from the other reviewers.

Line 1117: 'as in Fig 5K', instead of 4K?

Yes, thank you for catching that.

In Fig 5, you always relate Sox2 expression with SMAD4 levels, but what about SMAD4 integral? Does the decay in Sox2 expression also correlate with the integral of BMP signaling?

By design, SMAD4 signaling level was almost constant over time in these experiments, as in Fig. 4I. With a constant level for a fixed amount of time, integral and level are proportional. In this case, a graph for SOX2 rate of change versus SMAD4

signaling level is identical to one of SOX2 level versus SMAD4 signaling integral except with different units on the axes. We understand the confusion and have now used SMAD4 integral instead of level whenever appropriate.

Line 525: reference in Fig?

We have added a figure reference.

Line 559: SI Fig 4R, right?

Yes, now SI Fig. 4V.

In Fig 5L, M, over expression of Sox2 is found to reduce differentiation to amnion-like fate in response to BMP. Does the over expression of SOX2 lead to a change in the integral threshold for amnion cell fate specification?

That is a great question. Yes, that would be the prediction, however, we were unable to quantitatively relate total SOX2 over time to amnion-like fate since only one allele of endogenous SOX2 was GFP tagged and the other allele and the overexpressed SOX2 were not. We hope to pursue this further in the future, but given the amount of data in this manuscript, we consider tracking SOX2 under different conditions beyond the scope of the current work.

References

1. Etoc, F., Metzger, J., Ruzo, A., Kirst, C., Yoney, A., Ozair, M.Z., Brivanlou, A.H., and Siggia, E.D. (2016). A Balance between Secreted Inhibitors and Edge Sensing Controls Gastruloid Self-Organization. *Dev. Cell* 39, 302–315. 10.1016/j.devcel.2016.09.016.
2. Nemashkalo, A., Ruzo, A., Heemskerk, I., and Warmflash, A. (2017). Morphogen and community effects determine cell fates in response to BMP4 signaling in human embryonic stem cells. *Development*. 10.1242/dev.153239.
3. Yoney, A., Etoc, F., Ruzo, A., Carroll, T., Metzger, J.J., Martyn, I., Li, S., Kirst, C., Siggia, E.D., and Brivanlou, A.H. (2018). WNT signaling memory is required for ACTIVIN to function as a morphogen in human gastruloids. *Elife* 7. 10.7554/eLife.38279.
4. Heemskerk, I., Burt, K., Miller, M., Chhabra, S., Guerra, M.C., Liu, L., and Warmflash, A. (2019). Rapid changes in morphogen concentration control self-organized patterning in human embryonic stem cells. *Elife* 8. 10.7554/eLife.40526.
5. Grönroos, E., Kingston, I.J., Ramachandran, A., Randall, R.A., Vizán, P., and Hill, C.S. (2012). Transforming growth factor β inhibits bone morphogenetic protein-induced transcription through novel phosphorylated Smad1/5-Smad3 complexes. *Mol Cell Biol* 32, 2904–2916. 10.1128/MCB.00231-12.
6. Chhabra, S., Liu, L., Goh, R., Kong, X., and Warmflash, A. (2019). Dissecting the dynamics of signaling events in the BMP, WNT, and NODAL cascade during self-organized fate patterning in human gastruloids. *PLoS Biol*. 17, e3000498. 10.1371/journal.pbio.3000498.
7. Shonibare, Z., Monavarian, M., O'Connell, K., Altomare, D., Shelton, A., Mehta, S., Jaskula-Sztul, R., Phaeton, R., Starr, M.D., Whitaker, R., et al. (2022). Reciprocal SOX2 regulation by SMAD1-SMAD3 is critical for anoikis resistance and metastasis in cancer. *Cell Rep* 40, 111066. 10.1016/j.celrep.2022.111066.
8. Domyan, E.T., Ferretti, E., Throckmorton, K., Mishina, Y., Nicolis, S.K., and Sun, X. (2011). Signaling through BMP receptors promotes respiratory identity in the foregut via repression of Sox2. *Development* 138, 971–981. 10.1242/dev.053694.
9. Greber, B., Lehrach, H., and Adjaye, J. (2008). Control of early fate decisions in human ES cells by distinct states of TGFbeta pathway activity. *Stem Cells Dev* 17, 1065–1077. 10.1089/scd.2008.0035.
10. Liu, L., Michowski, W., Inuzuka, H., Shimizu, K., Nihira, N.T., Chick, J.M., Li, N., Geng, Y., Meng, A.Y., Ordureau, A., et al. (2017). G1 cyclins link proliferation, pluripotency and differentiation of embryonic stem cells. *Nat Cell Biol* 19, 177–188. 10.1038/ncb3474.
11. Streibinger, D., Deluz, C., Friman, E.T., Govindan, S., Alber, A.B., and Suter, D.M. (2019). Endogenous fluctuations of OCT4 and SOX2 bias pluripotent cell fate decisions. *Mol Syst Biol* 15, e9002. 10.15252/msb.20199002.
12. Alber, A.B., Paquet, E.R., Biserni, M., Naef, F., and Suter, D.M. (2018). Single Live Cell Monitoring of Protein Turnover Reveals Intercellular Variability and Cell-Cycle Dependence of Degradation Rates. *Mol Cell* 71, 1079-1091.e9. 10.1016/j.molcel.2018.07.023.
13. Alon, U. (2006). *An Introduction to Systems Biology: Design Principles of Biological Circuits* 1st edition. (Chapman and Hall/CRC).
14. Boyer, L.A., Lee, T.I., Cole, M.F., Johnstone, S.E., Levine, S.S., Zucker, J.P., Guenther, M.G., Kumar, R.M., Murray, H.L., Jenner, R.G., et al. (2005). Core transcriptional regulatory circuitry in human embryonic stem cells. *Cell* 122, 947–956. 10.1016/j.cell.2005.08.020.
15. Simon, C.S., Rahman, S., Raina, D., Schröter, C., and Hadjantonakis, A.-K. (2020). Live Visualization of ERK Activity in the Mouse Blastocyst Reveals Lineage-Specific Signaling Dynamics. *Dev. Cell* 55, 341-353.e5. 10.1016/j.devcel.2020.09.030.
16. Deathridge, J., Antolović, V., Parsons, M., and Chubb, J.R. (2019). Live imaging of ERK signalling dynamics in

differentiating mouse embryonic stem cells. *Development* 146, dev172940. 10.1242/dev.172940.

17. Petkova, M.D., Tkačik, G., Bialek, W., Wieschaus, E.F., and Gregor, T. (2019). Optimal Decoding of Cellular Identities in a Genetic Network. *Cell* 176, 844-855.e15. 10.1016/j.cell.2019.01.007.

18. Greenfeld, H., Lin, J., and Mullins, M.C. (2021). The BMP signaling gradient is interpreted through concentration thresholds in dorsal-ventral axial patterning. *PLoS Biol.* 19, e3001059. 10.1371/journal.pbio.3001059.

19. Lambert, S.A., Jolma, A., Campitelli, L.F., Das, P.K., Yin, Y., Albu, M., Chen, X., Taipale, J., Hughes, T.R., and Weirauch, M.T. (2018). The Human Transcription Factors. *Cell* 175, 598–599. 10.1016/j.cell.2018.09.045.

REVIEWERS' COMMENTS

Reviewer #1 (Remarks to the Author):

In their revised manuscript, Teague et al. have appropriately addressed all reviewers' comments, but please clarify in the Methods section how the half-life of Sox2 was determined from the FRAP data. It looks like $dc/dt = k_1 - k_2c$ with a normalized $c(t) = ae^{-(k_2t)} + k_1/k_2$ was used; but given the new inclusion of Sox2 autoregulation, something like $dc/dt = c - k_2c$ with $c(t) = ae^{-(t-k_2t)}$ or $dc/dt = c^4 - k_2c$ with $c(t) = (k_2)^{1/3} / (1 - e^{3k_2(a+t)})^{1/3}$ or their normalized versions may be more appropriate.

Reviewer #2 (Remarks to the Author):

The authors have revised the paper to address my original comments and concerns, and it is certainly now improved. I have no further issues.

Reviewer #3 (Remarks to the Author):

The authors have addressed all my concerns and have improved the manuscript. I have no further requests.

RESPONSE TO REVIEWERS

Reviewer comments are *italicized* with our response below.

Reviewer #1:

In their revised manuscript, Teague et al. have appropriately addressed all reviewers' comments, but please clarify in the Methods section how the half-life of Sox2 was determined from the FRAP data. It looks like $dc/dt=k_1-k_2c$ with a normalized $c(t)=ae^{-(k_2t)}+k_1/k_2$ was used; but given the new inclusion of Sox2 autoregulation, something like $dc/dt=c-k_2c$ with $c(t)=ae^{-(t-k_2t)}$ or $dc/dt=c^4-k_2c$ with $c(t)=(k_2)^{1/3}/(1-e^{3k_2(a+t)})^{1/3}$ or their normalized versions may be more appropriate.

Thank you for this question as it is important to consider the impact of feedback from SOX2 on the response curve. In our fluorescence recovery after photobleaching (FRAP) experiments we measured fluorescence recovery from protein turnover while total SOX2 remained constant. Thus, any non-linear production term like c^4 is constant and we still have simple exponential recovery with a timescale set by protein half-life. We have a description of how these measurements were made to the Methods.

Reviewer #2:

The authors have revised the paper to address my original comments and concerns, and it is certainly now improved. I have no further issues.

Reviewer #3:

The authors have addressed all my concerns and have improved the manuscript. I have no further requests.